# Mechanistic insights into photochemical nickel-catalyzed cross-couplings enabled by energy transfer

Rajesh Kancherla [1,4], Krishnamoorthy Muralirajan[1,4], Bholanath Maity [1,4], Safakath Karuthedath[2], Gadde Sathish Kumar[1], Frédéric Laquai [2], Luigi Cavallo [1] & Magnus Rueping [1,3 ✉]

Various methods that use a photocatalyst for electron transfer between an organic substrate and a transition metal catalyst have been established. While triplet sensitization of organic substrates via energy transfer from photocatalysts has been demonstrated, the sensitization of transition metal catalysts is still in its infancy. Here, we describe the selective alkylation of C($sp^3$)–H bonds via triplet sensitization of nickel catalytic intermediates with a thorough elucidation of its reaction mechanism. Exergonic Dexter energy transfer from an iridium photosensitizer promotes the nickel catalyst to the triplet state, thus enabling C–H functionalization via the release of bromine radical. Computational studies and transient absorption experiments support that the reaction proceeds via the formation of triplet states of the organometallic nickel catalyst by energy transfer.

[1] KAUST Catalysis Center (KCC), Physical Science and Engineering Division, King Abdullah University of Science and Technology (KAUST), Thuwal 23955-6900, Kingdom of Saudi Arabia. [2] KAUST Solar Center (KSC), Physical Science and Engineering Division, King Abdullah University of Science and Technology (KAUST), Thuwal 23955-6900, Kingdom of Saudi Arabia. [3] Institute for Experimental Molecular Imaging (ExMI), RWTH Aachen, University Clinic, Forckenbeckstr. 55, Aachen 52074, Germany. [4] These authors contributed equally: Rajesh Kancherla, Krishnamoorthy Muralirajan, Bholanath Maity. ✉email: magnus.rueping@kaust.edu.sa

In the last decade, the field of visible light-mediated photocatalysis has gained great attention, and cross-coupling yields can be improved by employing radicals created by electron and energy transfer processes[1,2]. Although photocatalytic processes involving electron transfer (ET) that depend on the redox properties of an excited-state molecule have been well established, reports of photocatalysis via energy transfer (EnT) are limited. Taking advantage of EnT processes, organic reactions such as cyclization[3–7], double bond isomerization[8–12], deracemisation[13,14], and bond dissociation[15–17] have been developed by the direct triplet sensitization of organic reactants. Surprisingly, processes involving the sensitization of organometallic complexes via EnT, specifically in the realm of carbon-carbon (C–C) bond formation, have remained comparably underdeveloped. Recently, advances were reported by Molander[18], Shibasaki[19], and our group[20] realizing the C($sp^3$)–H arylation, vinylation, and acylation by the triplet-state sensitization of an organometallic Ni(II)-complex. Whilst energy transfer for the homolysis of the Ni-halogen bond was invoked in these publications[18–20], reductive elimination from ground state Ni(II) complexes was suggested, which appears unlikely. In this context, Macmillan and coworkers have developed the C–O and C–N bond formation protocols and demonstrated that the unfavorable ground state reductive elimination from Ni(II) could be circumvented through energy transfer from an iridium photosensitizer to Ni(II)[21–23]. However, detailed mechanistic investigations of nickel catalyzed C–C cross-coupling reactions through energy transfer are less explored. Generally, in nickel catalysis [Ni(II)RX (R = aryl or alkyl; X = halogen)], achieving C($sp^3$)–C($sp^2$) and C($sp^3$)–C($sp^3$) cross-coupling reactions without using a photosensitizer (PS) is quite challenging since the Ni(II)–Br σ* orbital cannot be easily populated by the direct irradiation of nickel complexes. Therefore, using a PS with a sufficiently high triplet-state energy facilitates the activation of these nickel complexes by promoting an electron into the Ni(II)–Br σ* orbital via EnT, thus allowing cross-couplings to occur (Fig. 1). The EnT from the PS to the Ni(II) complex can follow two different mechanisms, namely, Förster or Dexter energy transfer. In the Förster mechanism[24], a non-radiative dipole-dipole interaction between the excited-state *PS and the ground-state Ni(II)–Br leads to an excited singlet state of

Ni(II)–Br (Fig. 1). On the other hand, in Dexter EnT[25,26], the simultaneous intermolecular exchange of two electrons between *PS and Ni(II)–Br leads to an excited triplet state of Ni(II)–Br (Fig. 1). A general overview of the EnT processes has been presented previously[27,28]. However, an in-depth understanding of the triplet sensitization of organometallic complexes is required to guide future developments in excited-state metal catalysis.

In this work, we describe the selective alkylation of α-oxy C($sp^3$)–H bonds by the direct coupling of ethers with alkyl bromides by excited-state nickel catalysis. Triplet sensitization of the organometallic Ni-complex, the photophysics, and the mechanism of photosensitised nickel excited state catalysis is studied by experimental investigations, computational calculations, and transient spectroscopic measurements.

## Results and discussion

**Development of C($sp^3$)–H alkylation.** Demonstrating the viability of EnT processes, we report a method for the direct cross-coupling of α-oxy C($sp^3$)–H bonds with alkyl bromides to give C($sp^3$)–C($sp^3$) coupled products using a sensitized nickel catalyst with both experimental and computational support (Fig. 2c). In this context, MacMillan and coworkers have reported selective C($sp^3$)–alkylation by polarity-matched hydrogen atom transfer (HAT) using a triple-catalytic combination of [Ir], [Ni], and a HAT catalyst (Fig. 2a)[29]. Recently, Paixão and König reported the C($sp^3$)–C($sp^3$) cross-coupling of alkyl bromides and chlorides with ethers using 4-CzIPN and Ni(II) acetylacetonate, where a single-electron transfer (SET) pathway was suggested (Fig. 2b)[30]. In our studies, we found that neither of the above mechanisms can be possible due to the absence of HAT reagent and the mismatch of redox potentials. Thus, we here report a full investigation of a photochemical nickel catalyzed C($sp^3$)–alkylation cross-coupling of ethers and alkyl bromides employing a combined experimental, computational and spectroscopic study.

**Reaction optimization.** Our initial test reaction between (3-bromopropyl)benzene and tetrahydrofuran (THF) using 2 mol% Ir[dF(CF$_3$)ppy]$_2$(dtbbpy)PF$_6$ (PS1), 5 mol% NiCl$_2$·glyme, 6 mol% 4,4′-di-*tert*-butyl-2,2′-bipyridine (4,4′-dtbbpy) and 2 equiv. of K$_2$CO$_3$ gave cross-coupled product 1 in 36% yield after visible light irradiation at RT. However, optimizing various parameters,

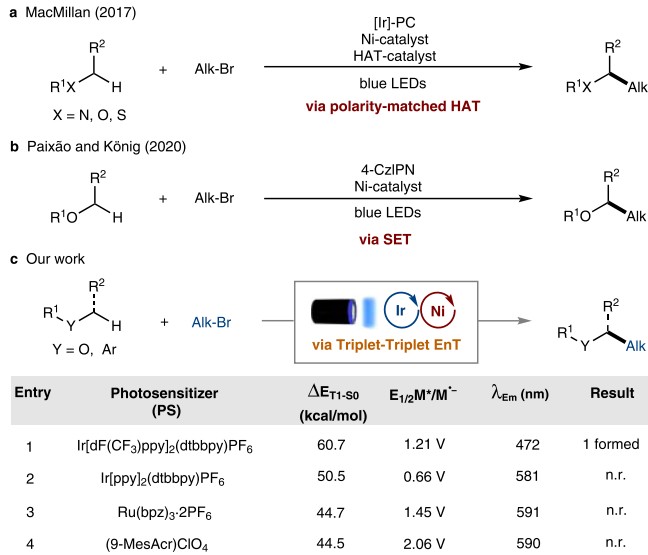

**Fig. 1 Förster and Dexter EnT for organometallic Ni-catalyzed C($sp^3$)–H arylation and alkylation.** PS photosensitizer, EnT energy transfer.

**Fig. 2 Photochemical/Ni-catalyzed C($sp^3$)–C($sp^3$) cross-coupling.**
**a** Report by MacMillan via polarity-matched HAT process. **b** Report by Paixão and König via SET process. **c** Our work via energy transfer process.

| Entry | Photosensitizer (PS) | ΔE$_{T1-S0}$ (kcal/mol) | E$_{1/2}$M*/M·⁻ | λ$_{Em}$ (nm) | Result |
|---|---|---|---|---|---|
| 1 | Ir[dF(CF$_3$)ppy]$_2$(dtbbpy)PF$_6$ | 60.7 | 1.21 V | 472 | 1 formed |
| 2 | Ir[ppy]$_2$(dtbbpy)PF$_6$ | 50.5 | 0.66 V | 581 | n.r. |
| 3 | Ru(bpz)$_3$·2PF$_6$ | 44.7 | 1.45 V | 591 | n.r. |
| 4 | (9-MesAcr)ClO$_4$ | 44.5 | 2.06 V | 590 | n.r. |

**Table 1 Optimization of reaction conditions[a].**

Ph$\diagdown\diagdown$Br + [tetrahydrofuran] $\xrightarrow[\text{K}_2\text{CO}_3 \text{ (2 equiv.), blue LEDs, RT}]{\begin{array}{c}\text{Ir[dF(CF}_3)\text{ppy]}_2\text{(dtbbpy)PF}_6 \text{ (2 mol\%)}\\ \text{NiCl}_2.\text{glyme (5 mol\%)}\\ \text{4,4'-dOMe-bpy (5.5 mol\%)}\end{array}}$ [product] Ph **1**

| Entry | Change in standard reaction conditions | Yield (1, %)[b] |
|---|---|---|
| 1 | NiCl$_2$.glyme (5 mol%), 4,4'-dtbbpy (6 mol%) | 36 |
| 2 | None | 54 |
| 3 | Ni(cod)$_2$ (5 mol%), 4,4'-dOMe-bpy (5.5 mol%) | 77 |
| 4 | Ni(cod)$_2$ (10 mol%), 4,4'-dOMe-bpy (11 mol%) | 75 |
| 5 | H$_2$O (10 equiv.) as additive | 73 |
| 6 | 1 mol% Ir[dF(CF$_3$)ppy]$_2$(dtbbpy)PF$_6$, Ni(cod)$_2$ (5 mol%) | 74 |
| 7 | Ir[ppy]$_2$(dtbbpy)PF$_6$ instead of PS1 | 0 |
| 8 | Ru(bpz)$_3$·2PF$_6$ instead of PS1 | 0 |
| 9 | (9-MesAcr)ClO$_4$ instead of PS1 | 0 |
| 10 | 4-CzIPN (2 mol%) instead of PS1, Ni(cod)$_2$ (5 mol%) | 30 |
| 11 | without [Ir] photosensitizer | 0 |
| 12 | without [Ni] catalyst | 0 |
| 13 | without Light source | 0 |
| 14 | without base | trace |
| 15 | without fan cooling (around 50 °C) | 51 |
| 16 | (3-chloropropyl)benzene instead of alkyl bromide | 0 |

[a]Standard conditions: Alkyl bromide (0.1 mmol), THF (0.05 M, 2 mL), NiCl$_2$.glyme (5 mol%), 4,4'-dOMe-bpy (4,4'-dimethoxy-2,2'-bipyridyl) (5.5 mol%), Ir[dF(CF$_3$)ppy]$_2$(dtbbpy)PF$_6$ (2 mol%), K$_2$CO$_3$ (2 equiv.), 34 W blue LEDs, Ar, 48 h, room temperature.
[b]Yield determined by GC. Emission maximum of the light source used is 425 nm.

including changing the transition metal (TM) catalyst and ligand to (dOMe-bpy)Ni(cod), boosted the yield to 77% (Table 1). Control experiments demonstrated that all the individual components, e.g., [Ir], [Ni], base, and visible light, are necessary for the reaction to proceed. Next, photosensitizers including strongly oxidizing Ru(bpz)$_3$.2PF$_6$ (PS3) and (9-MesAcr)ClO$_4$ (P4), were paired with nickel in a standard reaction to rule out the possibility of the reaction proceeding through a single electron transfer (SET) via oxidation of the Ni(II) complex to Ni(III), allowing the Ni(III) halide to catalyze the C–H functionalization through a halogen photoelimination[31–34]. However, these reactions did not yield any cross-coupled product, suggesting that a mechanism involving the oxidation of Ni(II) to Ni(III) might not be operative (Fig. 2c, and Table 1). Also, electron transfer involving the oxidation of THF by a photocatalyst[29,35,36] is not plausible due to its significantly higher oxidation potential (oxidation onset potential of THF is E = +1.75 V vs SCE)[33,37], which makes its oxidation by Ir[dF(CF$_3$)ppy]2(dtbbpy)PF$_6$ (Ir(III)*/Ir(II) = +1.21 V vs SCE) or other photocatalysts unlikely, owing to their lower reduction potentials. Based on these observations and DFT studies (vide infra), an alternative EnT mechanism is proposed.

**Scope of substrates**. Next, we started to explore the scope of this C(sp$^3$)–H alkylation using the optimized reaction conditions (Fig. 3), providing products in good to moderate yields (1–28). For example, alkyl halides with different functional groups, such as amides (4), esters (5–8), and nitriles (14), gave the corresponding products in good yields. Compound 5 gave less yield compared to compound 6 since the Ni(II) intermediate undergoes β-hydride elimination resulting in the formation of the olefination side product. A branched alkyl halide successfully gave the corresponding cross-coupled product 10 in 52% yield. A dioxolane derivative was also found to be reactive and gave aldehyde 13 in 62% yield upon deprotection. Notably, cyclic systems were also found to be suitable substrates (15: 60%, 16: 52%), allowing secondary–secondary bond formation between dissimilar cyclic systems, which has been viewed as a challenge in cross-coupling methodology. In addition, bulky 17β-(bromomethyl)-3β-methoxy-5-androstene also gave cross-

coupled product 17, displaying the synthetic ability of this transformation. The diminished reactivity, in this case, can be due to the bulkiness of the alkyl bromide, which is resulting in the formation of a hydrogenation side product rather than a cross-coupled product. Subsequently, phenethylbromides with electron-donating and withdrawing groups on the phenyl ring provided the corresponding cross-coupled products in good yields (18–22). Furthermore, dialkylated product 23 was obtained when 1,6-dibromohexane was reacted with THF under the optimized reaction conditions. Importantly, in the case of 1-bromo-6-chlorohexane, alkylation took place selectively at the C–Br site, leaving the C–Cl bond unreacted, providing an opportunity to further functionalize the reaction product (24, 61%). Next, the scope of the alkylation protocol was examined concerning different C(sp$^3$)–H bonds. Ethereal solvents such as d$_8$-THF and 1,4-dioxane reacted under the optimized reaction conditions to give products 25–27 in 24–57% yields. The greater reactivity of THF compared to other ethereal solvents can be attributed to stereoelectronic factors. In the case of 1,4-dioxane, the diminished reactivity might be mainly due to the inductive effect making the C–H bond less hydridic and thereby less prone to abstraction by the electrophilic radical[38,39]. Additionally, a non-ethereal substrate toluene was to be found reactive and gave the cross-coupled product 28. In this case an outer-sphere C(sp$^3$)–H activation may be operative which is giving a stable benzylic radical. Once the benzylic radical is formed it can either attack the alkyl-Ni(I) intermediate (E$^D$, ΔG = −15.8 kcal/mol) or dimerizes (ΔG = −24.4 kcal/mol). Since the dimerization of the benzylic radical is more favored over the addition to alkyl-Ni(I) intermediate (E$^D$), the reaction with toluene resulted in a lower yield compared to THF. We also tried to use THF in equivalent amounts, however, homocoupling of alkyl halide was obtained as a major product. Therefore, we used THF as solvent in order to suppress the homocoupling and to promote the cross-coupling reaction.

**Computational study**. Detailed DFT calculations were performed to define the reaction mechanism and elucidate the nature of triplet sensitization. The overall calculated reaction pathway (Fig. 4) was divided into three sections: (i) oxidative addition,

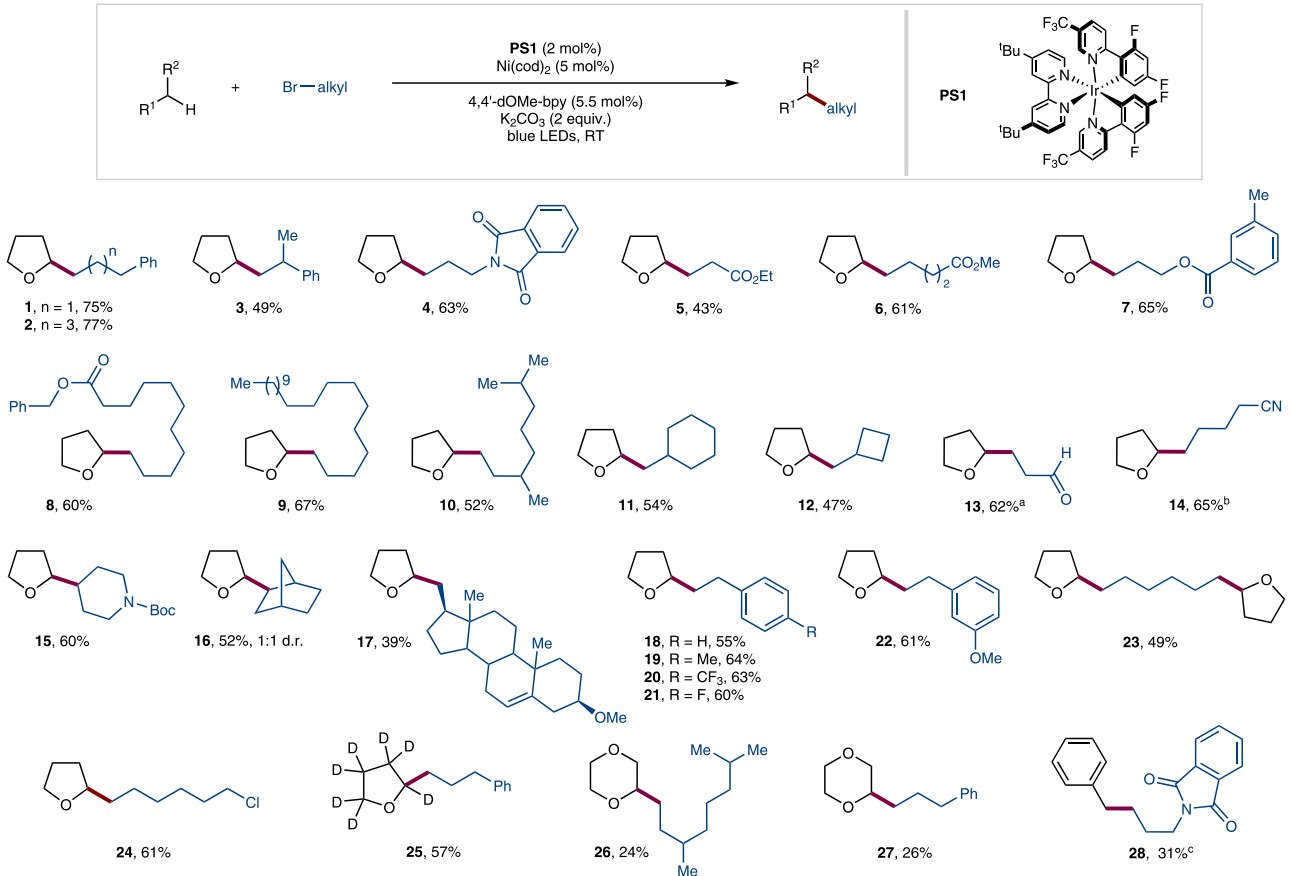

**Fig. 3 Substrate scope.** Reaction conditions: alkyl bromide (0.2 mmol), C–H coupling partner as solvent (0.05 M, 4 mL), Ir[dF(CF$_3$)ppy]$_2$(dtbbpy)PF$_6$ (PS1, 2 mol%), Ni(cod)$_2$ (5 mol%), 4,4′-dOMe-bpy (5.5 mol%), K$_2$CO$_3$ (2 equiv.), 34 W blue LEDs, Ar, 48 h, room temperature. [a]The reaction was performed using 2-(2-bromoethyl)-1,3-dioxolane and the final aldehyde product was obtained upon deprotection. [b]Isolated in a 5:1 ratio with the homo-coupling product of alkyl halide. [c]10 equiv. of alkyl component, benzene (0.125 M), NiCl$_2$ glyme (10 mol%), 4,4′-dtbbpy (11 mol%), NaHCO$_3$ (2 equiv.), 34 W x 2 blue LEDs, 96 h. See Supplementary Methods 3 for more details.

(ii) EnT followed by hydrogen atom transfer (HAT), and (iii) reductive elimination. For the oxidative addition of the alkyl bromide to the Ni(0) complex, we investigated two possible routes: first, the Ni complex in the singlet electronic state (orange line in Fig. 4) and second, the Ni complex in the triplet state (blue line in Fig. 4). Both routes start with the replacement of the cod ligand in the A$^S$ or A$^T$ complexes (the latter is only 4.0 kcal/mol higher in energy) by alkyl bromide 1$_R$. Those steps generate intermediates B$^S$ and B$^T$, endergonic by 27.8 and 16.6 kcal/mol, respectively. Alkyl bromide activation from B$^S$ follows an S$_N$2-type mechanism via the transition state [B-C]$^S$ with the liberation of Br$^-$ from 1$_R$ and its subsequent coordination to the Ni center to arrive at Ni(II) complex D$^S$. We note that a classic three-center transition state for oxidative addition for the simultaneous formation of Ni–Br, and Ni–C bonds could not be located, indicating this route is unfavorable[40]. The activation of B$^T$ starts with a SET step via transition state [B-C]$^T$ and then proceeds through biradical species C$^T$ before converging into triplet Ni(II) complex D$^T$, which can finally relax to singlet Ni(II) complex D$^S$. According to our calculations, oxidative addition along the triplet pathway is favored, as transition state [B-C]$^T$ is 16.4 kcal/mol lower in energy than transition state [B-C]$^S$. The product of the oxidative addition step, singlet Ni(II) intermediate D$^S$, is 22.2 kcal/mol lower in energy than starting A$^S$ and 1$_R$ species.

The next step is promoted by Ir(III) photosensitizer PS1, which has a long-lived triplet excited-state *PS1 ($\tau_0 = 1.865 \pm 0.003$ μs)

that activates D$^S$ either by ET, EnT, or both. Within the ET mechanism, D$^S$ is oxidized by triplet Ir(III) species *PS1 to cationic Ni(III) species H$^D$ in the doublet state, and this step is exergonic by 21.4 kcal/mol (Supplementary Fig. 15). However, the high-energy barrier for the following C($sp^3$)–H activation step ($\Delta G^{\ddagger} = 45.9$ kcal/mol) prevents further progress along this pathway. This suggests that the quenching of the photoluminescence of *Ir(III) through interaction with Ni(II) to result cross-coupled product is due to EnT. A similar conclusion was proposed in the context of Fe catalysis in the presence of an Ir photocatalyst[40]. Within the EnT mechanism, the interaction between *PS1 and D$^S$ should trigger the breaking of the Ni–Br bond by the promotion of one electron from an occupied molecular orbital to the Ni–Br σ* orbital. This can occur via the Dexter energy transfer mechanism[25,26], which corresponds to the spontaneous mutual exchange of electrons between *PS1 and D$^S$, resulting in PS1 and *D$^T$, with the latter corresponding to excited Ni(II) in the triplet state (Fig. 4, details in Supplementary Figs. 17, 18). A Förster resonance energy transfer mechanism can be ruled out in our system considering the basic spin conversion rules[41].

To investigate the above possible pathway and to obtain electronic information of higher energy triplet states, we performed TD-DFT calculations (Supplementary Methods 14). Analysis of the orbitals contributing to the dominant electronic transitions of D$^S$ in the UV-Vis region revealed that all the excitations correspond to transitions of electrons from occupied Ni orbitals to the LUMO, LUMO + 1, and LUMO + 2, which

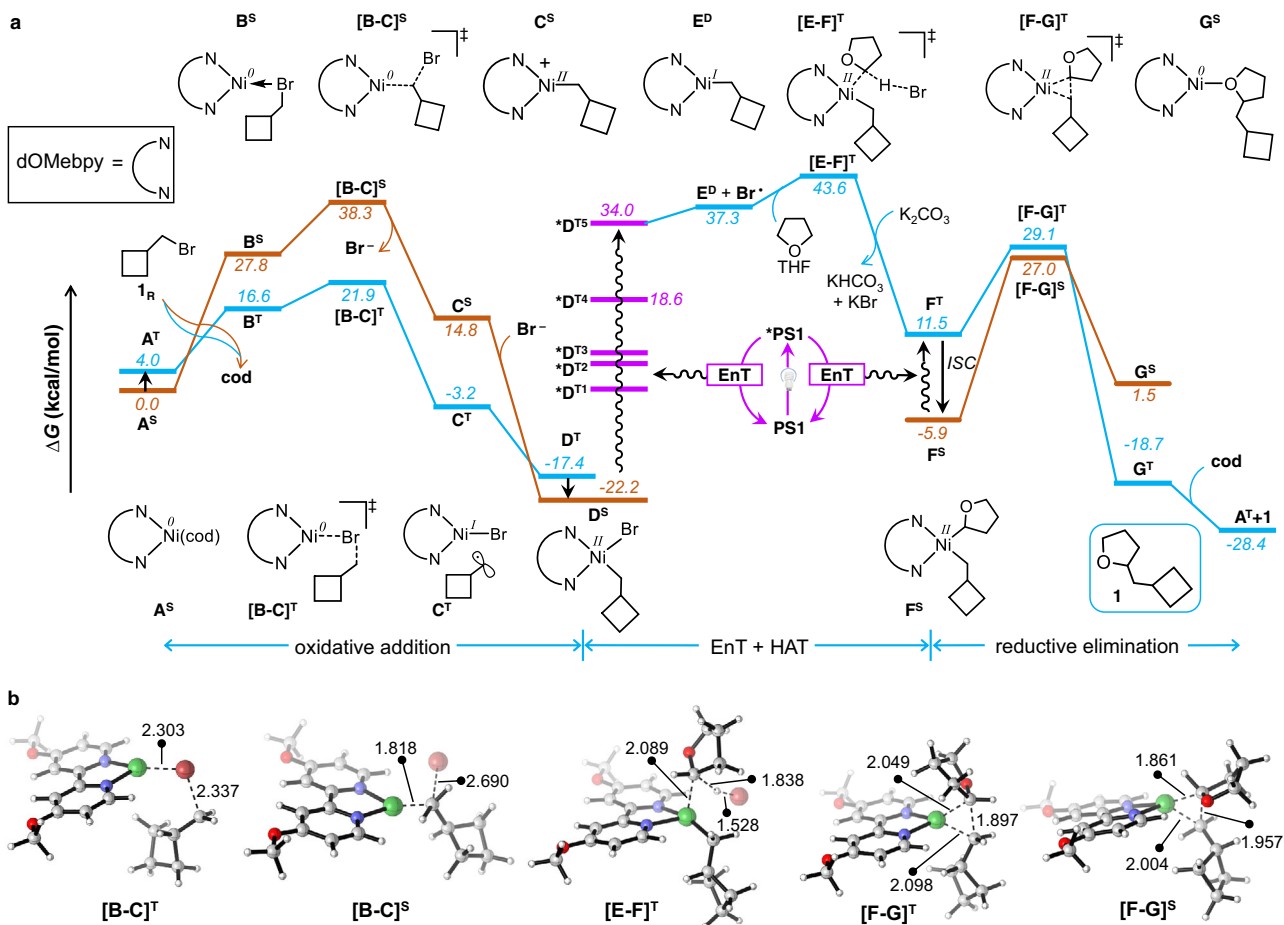

**Fig. 4 DFT study. a** Computed energy profile for the nickel-catalyzed direct C($sp^3$)–H alkylation. Free energies in solution (THF) at the M06(SMD)/Def2-TZVPP//M06/Def2-TZVP(Ni,Ir)/Def2-SVP(nonmetals) level are displayed. S, T, and D superscripts indicate singlet, triplet, and doublet electronic states, respectively. The HAT step leads to the formation of HBr which is readily transformed into KBr and KHCO$_3$ in presence of K$_2$CO$_3$. **b** Optimized geometries of all transition states. Bond lengths are in Å.

represent π* molecular orbitals located on the dOMe-bpy ligand (Supplementary Table 5, and Supplementary Fig. 16)[42]. These excitations are not effective in promoting the reactivity of the Ni(II) complex towards active bromine radical formation since the electron is not promoted to the Ni–Br σ* orbital (LUMO + 3), as this would be a Laporte-forbidden d–d* transition. Confirming this finding experimentally, (dtbbpy)Ni(II)-alkyl bromide prepared in situ displayed broad absorption features with λ$_{max}$ values of 470 and 283 nm due to metal-to-ligand charge transfer ($^1$MLCT) and a ligand-centered π → π* transition, respectively (Supplementary Fig. 1). An ISC (inter-system crossing) of direct photoexcited state $^1$MLCT leads to $^3$MLCT type triplet, which eventually undergoes IC (internal conversion) to the most stable $^3$d-d state D$^T$ which is in agreement to the recent report by Doyle and coworkers[43]. However, in the current C($sp^3$)–H alkylation reaction the $^3$d-d state D$^T$ is unproductive, since it is not the source of bromide radical. Irradiating the in situ-generated Ni(II)-alkyl bromide complex with visible and ultraviolet light in the absence of PS1 either resulted in no cross-coupled product (with visible light) or trace product (with UV light, 300 nm), supporting the above conclusions.

Conversely, calculations revealed that the lowest singlet-triplet energy gap for PS1 (ΔG$_{T1-S0}$ = 60.7 kcal/mol) is larger than lowest five triplet excited states (T1 to T5) of D$^S$ (Supplementary Fig. 18). Analysis of the molecular orbitals of D$^S$ indicates that LUMO + 3 is involved in the T5 excited state, *D$^{T5}$, with a singlet-triplet energy gap, ΔG$_{T5-S0}$, of 56.2 kcal/mol. The T5 state

correspond to electron population at Ni–Br σ* bond, which is mostly contributed by Ni-d$_{x^2-y^2}$ orbital (Supplementary Fig. 16). Therefore, the T5 state readily undergoes Ni–Br homolysis to generate active bromine radical. This suggests that the promotion of one electron to this orbital via the Dexter EnT mechanism can induce homolytic cleavage of the Ni–Br bond to afford E$^D$, with the release of active bromine radical (Fig. 4). Of course, the Dexter EnT transfer can occur from *PS1 to any of the first 5 triplet excited states of D$^S$, with only that leading to T5 being effective in catalysis (Supplementary Discussion 5). This hypothesis suggests that photosensitizers having singlet-triplet energy gaps smaller than 56.2 kcal/mol should be less favored to activate the Ni(II) intermediate D$^S$[44]. Consistent with the experimental evidence, attempted cross-couplings in the presence of PS2, PS3, and PS4 (with singlet-triplet energy gaps of 50.5, 44.7, and 44.5 kcal/mol, respectively) showed no reactivity (Table 1, and Supplementary Fig. 18). Interestingly, reaction with 4-CzIPN with reasonably high triplet energy (E$_T$ = 58.3 kcal/mol, Table 1) gave the cross-coupled product in 30% yield[45]. Having clarified the EnT step, the second section of the reaction pathway ends with the HAT step, which consists of the bromine radical released from *D$^{T5}$ abstracting one of the α–H atoms of THF via triplet transition state [E-F]$^T$, which has an activation barrier of only 6.3 kcal/mol. The third and last section of the reaction pathway, reductive elimination from F$^T$ leading to G$^T$, occurs via transition state [F-G]$^T$ and has an activation barrier of 17.6 kcal/mol. Dissociation of product **1** from G$^T$ regenerates active species A$^T$,

which can then initiate another catalytic cycle. As an alternative, we explored reductive elimination from singlet and ground-state intermediate $F^S$, which can be formed by the decay of $F^T$. In this case, reductive elimination occurs via transition state $[F\text{-}G]^S$ and has a high activation barrier of 32.9 kcal/mol (Fig. 4). Therefore, if $F^T$ decays to $F^S$, the occurrence of reductive elimination requires excitation of $F^S$ back to $F^T$ by one of the EnT processes discussed above. Alternatively, the $F^S$ intermediate can be oxidized by the SET step before the reductive elimination step (Supplementary Discussion 2). In order to assess the number of photons involved in the reaction, the dependence of the reaction rate on light intensity was calculated by conducting the reactions under full and half intensity of the light irradiation at different time intervals. The results indicate that more than one photon is involved in the reaction mechanism (Supplementary Table 4).

**Spectroscopic studies.** To further clarify the reaction mechanism, steady-state Stern-Volmer luminescence quenching of *PS1 in the presence of different concentrations of Ni(II) alkyl bromide complex $D^S$ (prepared in situ) was investigated, and a linear correlation was found (Supplementary Fig. 4). A similar correlation was also revealed by time-resolved photoluminescence spectroscopy (TRPL), indicating excited-state quenching of *PS1 in the presence of different concentrations of $D^S$ (Fig. 5a, b). Both observations demonstrate that the quenching mechanism between the excited-state *PS1 and $D^S$ is dynamic and that there is no ground-state association between the photosensitizer and $D^S$ in the solution. Furthermore, the triplet-triplet EnT rate constant $k_{TTEnT}$ was determined by TRPL measurements. A $k_{TTEnT}$ of $(7.95 \pm 0.31) \times 10^9$ L mol$^{-1}$ s$^{-1}$ was determined from the linear

fit of the plot of the observed (measured) rate constant ($k_{obs}$) corrected by the ground-state recovery rate ($k_{GSR}$) of *PS1 (obtained in the absence of $D^S$) versus different concentrations of $D^S$ (Fig. 5c, and Supplementary Fig. 7). Since there is no sizeable intercept, we believe that the reverse TTEnT might not be operative[17,46,47].

To shed more light on the triplet energy transfer dynamics, we performed nanosecond transient absorption (ns-TA) pump-probe spectroscopy on PS1 in the presence of different concentrations of $D^S$ (Fig. 5d, h)[23,43,48]. Experimental details can be found in the Supplementary Information and from our previous publications[49,50]. Note that the positive signal in the spectra can be attributed to photoinduced excited-state absorption (ESA), and the negative signal represents ground state bleaching (GSB). Figure 5d shows the normalized ns-TA spectra of neat *PS1 with two broad peaks at 470 nm and 850 nm are observed, which we assigned to triplet-induced absorption of the photocatalyst as a consequence of MLCT/LC[51]. Non-normalized spectra are shown in the Supplementary Information (Supplementary Fig. 8). Further, on the addition of $D^S$, we still affirm that the band at 470 nm is triplet induced absorption of the photocatalyst for two reasons: 1) $D^S$ excited state absorption maxima is expected at 500 nm due to ligand-centered transition, and 2) they decay within 1 ns (Supplementary Fig. 9). In the absence of $D^S$, the $^3$MLCT state lifetime from TA is $1.865 \pm 0.003$ μs (Supplementary Fig. 10), which is in line with the phosphorescence lifetime of $1.967 \pm 0.002$ μs determined by TRPL (Supplementary Fig. 10). The $^3$MLCT state lifetime monotonically decreased upon the addition of the quencher, $D^S$. The transient absorption spectra observed after the addition of $D^S$ (0.2 mM) displayed a new excited-state absorption at

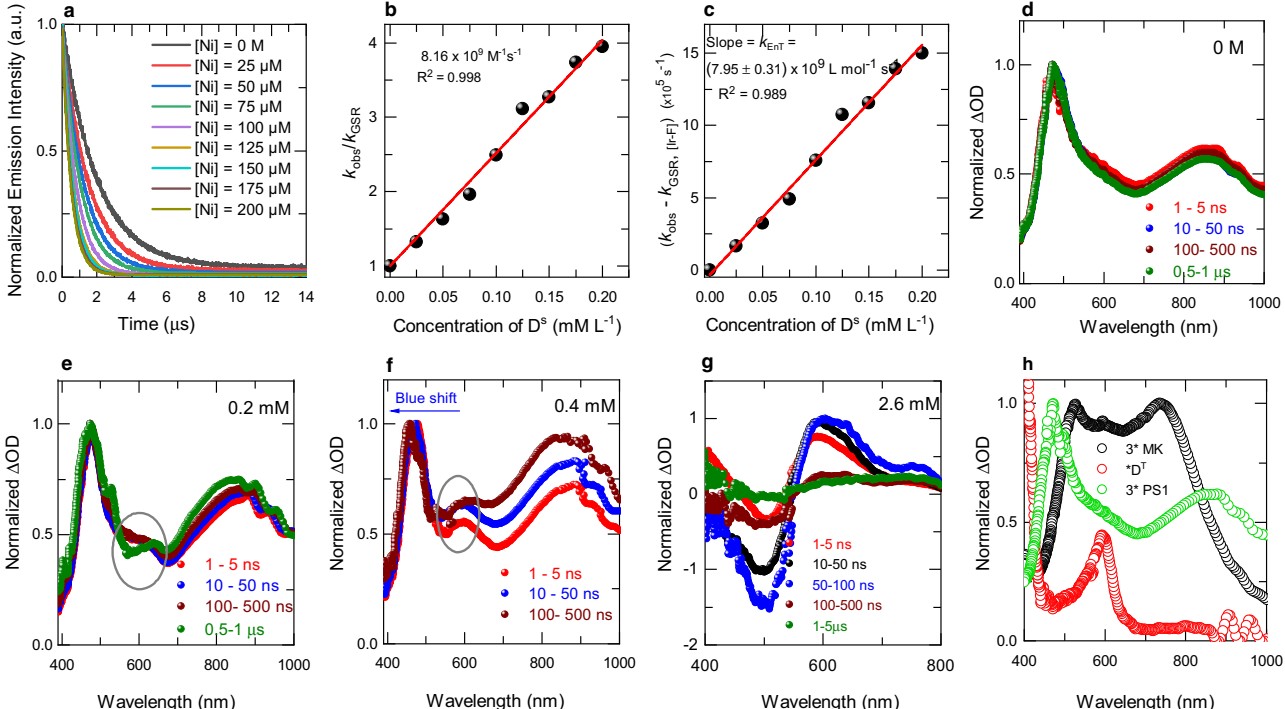

**Fig. 5 Excited-state dynamics. a** Reduction of the phosphorescence lifetime of *PS1 upon increasing the concentration of the quencher $D^S$. **b** Stern-Volmer plot of the phosphorescence lifetime quenching with increasing quencher concentration. **c** Plot of the observed rate constant ($k_{obs}$) of *PS1 corrected by the intrinsic ground state recovery rate ($k_{GSR}$) vs. different concentrations of $D^S$ to determine the triplet-triplet EnT rate constant ($k_{TTEnT}$). **d** Normalized ns-TA spectra of *PS1 (0.1 mM) in the absence of quencher $D^S$. **e, f** Normalized ns-TA spectra of a mixture of *PS1 (0.1 mM) and $D^S$ (0.2 and 0.4 mM) indicating the formation of *$D^T$ at 600 nm. **g** Normalized ns-TA spectra of a mixture of *PS1 (0.1 mM) and $D^S$ (2.6 mM), which shows complete quenching of the photocatalyst by $D^S$, resulting in a new ESA spectral feature at 400 and 600 nm; and a GSB at 500 nm. **h** Comparison of the decay associated states (DAS) of $^3$*(Michler's ketone) (0.1 mM; MK), neat *PS1 (0.1 mM), and the formation of triplet states of nickel (*$D^T$) by energy transfer from $^3$*MK.

600 nm (Fig. 5e). This excited-state absorption band at 600 nm becomes prominent with the increase in the concentration of quencher $D^S$ (Fig. 5f). At 2.6 mM of quencher $D^S$, bands at 470 nm and 850 nm corresponding to *PS1 completely disappeared, and two new excited-state absorption (ESA) bands at 400 and 600 nm were observed along with a ground-state bleach (GSB) at 500 nm (Fig. 5g). At low concentrations of the quencher, the band at 400 nm was covered by the *PS1 photoinduced signals; however, it became more pronounced at higher concentrations of the quencher and we assign these newly formed TA signals for *$D^T$ triplet state.

In case the reaction to proceeds via a Ni(III) intermediate by adding the quencher ($D^S$) to *PS1, an electron from Ni(II) enters into the empty $t_{2g}$ orbital of *PS1 which results in the formation of reduced PS1 (i.e. PS1$^{•−}$) with new transient spectral features and maxima at 400, 443, 499, and 530 nm[52]. Therefore, with the increase of the quencher ($D^S$) concentration, the ESA peaks (470 and 850 nm) corresponding to *PS1 should decay, and new TA signals corresponding to PS1$^{•−}$ and Ni(III) should appear. Overall, the quenching rate for these TA signals should be different for the ET pathway[53]. However, Fig. 5d–g, and Supplementary Fig. 8 clearly show that ESA peaks at 470 and 850 nm have disappeared with the concentration increase of $D^S$ and new transient peaks (ESA and GSB) appeared that are different from PS1$^{•−}$. Moreover, the quenching rate (450–480, and 775–880 nm) and the formation rate (590–650 nm) of the transient signals are identical (Fig. 6a) which clearly shows that the ET mechanism is not operative and an alternative EnT pathway is taking place. In the EnT pathway, spectroscopically, the mutual exchange of electrons between *PS1 and $D^S$ should result in the decay of ESA transient signals (470 and 850 nm, corresponding to reduced bipyridine ligand of *PS1), since it transfers that electron to the $T^5$ excited state of Ni(II). Simultaneously an electron from the ground state $D^S$ will be promoted to empty $t_{2g}$ orbital of *PS1 giving rise to ground-state PS1 and excited *$D^T$. Therefore the overall transient spectral changes involve the decay of ESA (470 and 850 nm), and the appearance of new transient signals corresponding to the excited *$D^T$. In addition, the excited *$D^T$ should have a long lifetime as the mutual exchange of electrons results in the formation of a

spin-flipped state. As expected, the ns-TA of *$D^T$ displayed a long-lived excited state with a weighted average lifetime of $\tau = 671$ ns (Supplementary Fig. 11) and we assign this newly formed long-lived excited state of nickel to be a spin-flipped triplet state. Besides PS1$^{•−}$ has a weighted average lifetime of $\tau = 86.71$ μs which is much higher than the lifetime of the new transient species ($\tau = 671$ ns) observed in our system, which further proves that PS1$^{•−}$ is not forming in our system[52]. Also, the TA bands at 450–480, and 775–880 nm were found to decay at the same rate with the generation of the new band formed at 590–650 nm (Fig. 6a) thus supporting that the quenching is occurring by the EnT mechanism. Further, to ensure the formation of *$D^T$ by triplet sensitization, a separate ns-TA on Michler's ketone (MK, 4,4'-bis(dimethylamino)benzophenone, with a singlet-triplet energy gap of 61.0 kcal/mol) was performed. Upon addition of the quencher ($D^S$), the same transient features at 400 and 600 nm (Fig. 5h) were observed, indicating that the state was formed by EnT from the photocatalyst. A separate catalytic reaction performed between THF and alkyl bromide using MK as photocatalyst which gave the $C(sp^3)$–$C(sp^3)$ cross-coupled product albeit in low yield supporting that the EnT mechanism is operative (Fig. 6c, and Supplementary Methods 10).

The latest work by Doyle[43], MacMillan and Scholes[23] group showed that neat Ni(II) aryl halide and Ni(II) aryl acetate complexes respectively have an excited state lifetime of $\tau = \sim 4$ ns (life-time of excited state Ni(II) in the absence of external photocatalyst). In order to verify the lifetime obtained in our case, we carried both the ps-TA and ns-TA spectroscopy on the Ni($^{t-Bu}$bpy) ($o$-Tol)Cl (Supplementary Methods 13) in the presence and the absence of the external photocatalyst PS1. In the absence of PS1, Ni($^{t-Bu}$bpy) ($o$-Tol)Cl has shown an excited state lifetime of $\tau = 1.3$ ns (Supplementary Fig. 12), similar to the result by the Doyle group[43]. Interestingly, the ps-TA spectra of a mixture of PS1 and Ni(II) ($o$-Tol)Cl showed a fast decay and then a slow but clear rise in the GSB band (Supplementary Fig. 13), indicating the formation of a very long-lived state. To quantify the decay dynamics of this state, we probed the kinetics evolution by ns-TA spectroscopy exciting at 355 nm (Supplementary Fig. 14). The decay concluded within about 20 ns but generated a new PA band at around 450 nm (Supplementary Fig. 14a) and

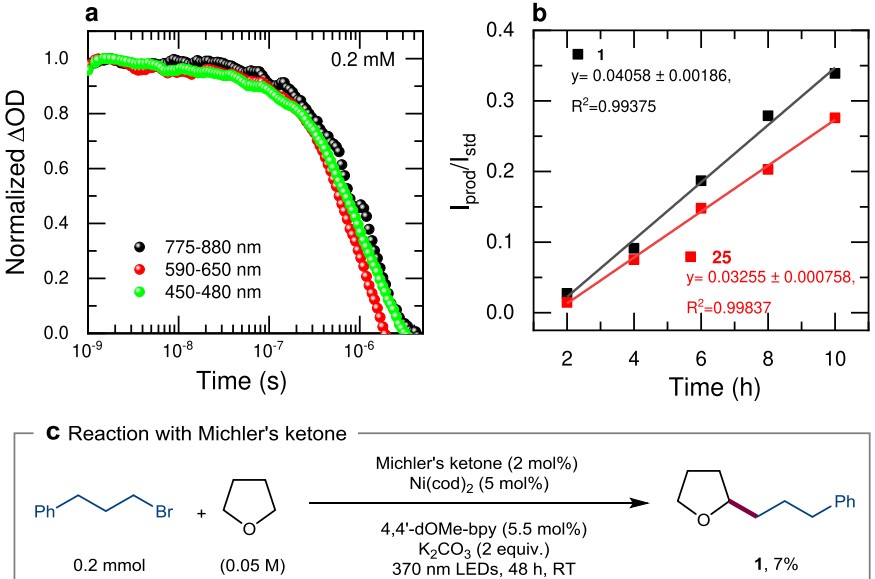

**Fig. 6 Mechanistic study. a** ns-TA kinetics of a mixture of *PS1 (0.1 mM) and $D^S$ (0.2 mM) indicating that the quenching rate and formation rate are identical. **b** Kinetic isotope effect using rate measurement. **1** is with undeuterated THF, and **25** is with deuterated THF substrate. **c** Control reaction using Michler's ketone as a photocatalyst to see the formation of the cross-coupled product via energy transfer from $^3$*MK.

550 nm, indicating the formation of a very long-lived state. The formation of a very long-lived state is also evidenced in the redshift of the GSB band. This state has a decay lifetime of 865 ns (Supplementary Fig. 14c) and very much in line with the observation in our material system.

Intermolecular competition experiments with a mixture of THF and $d_8$-THF (1:1) were carried out to understand the nature of the C–H functionalization. Under the optimized reaction conditions, the product distribution of **1** and **25** were in the ratio 1.7:1 (Supplementary Methods 9). Also, KIE was calculated using rate measurement which was found to be 1.2 (Fig. 6b). The small KIE observed in conjunction with the DFT studies (*vide supra*) reveals that the radical C–H abstraction is slightly endothermic with a transition state that closely resembles the product, and such a small KIE value indicates that the C–H bond breaking is not involved in the rate-determining step[54–56].

In conclusion, a protocol for the selective alkylation of α-oxy $C(sp^3)$–H bonds has been developed by the direct coupling of ethers and toluene with alkyl bromides by excited-state nickel catalysis. DFT and TD-DFT calculations point to a mechanism involving a Dexter triplet-triplet EnT from the *Ir(III) photo-sensitizer to the organometallic Ni(II) catalyst. The formation of the Ni(II) triplet state was further supported by nanosecond transient absorption spectroscopy, and the energy transfer rate constant was determined by time-resolved photoluminescence studies. The crucial C–H functionalization step is mediated by the bromine radical generated by the homolytic cleavage of the Ni–Br bond of the excited-state Ni(II) catalyst. The scope of the reaction was explored using a variety of alkyl halides, including secondary halides that reacted efficiently to enable secondary–secondary carbon bond formation. Furthermore, the study highlights that for photochemical/metal-catalyzed reactions next to SET pathways also energy transfer processes need to be considered. In summary, our combined experimental, computational studies, and detailed spectroscopic measurements allowed us to provide an insight into the photophysics and mechanism of photosensitised nickel excited state catalysis and will guide the further development of this exciting field of catalysis.

## Methods

### General procedure for photochemical nickel-catalyzed cross-coupling of THF and alkyl bromide to give C($sp^3$)–H alkylation product 1–28.

An oven-dried screw cap reaction tube and a 5 mL vial equipped with a PTFE-coated stir bar were brought into the $N_2$-filled glove box. Ni(cod)$_2$ (2.75 mg, 0.01 mmol, 5 mol%), 4,4′-dimethoxy-2,2′-bipyridyl (2.4 mg, 0.011 mmol, 5.5 mol%) and C($sp^3$)–H coupling partner (3 mL) was added in 5 mL vial and stirred well for 15 min to give a deep purple color solution (Mixture 1). The other reaction tube was charged with Ir[dF(CF$_3$)ppy]$_2$(dtbbpy)PF$_6$ (4.5 mg, 0.004 mmol, 2 mol%), K$_2$CO$_3$ (55 mg, 0.4 mmol, 2 equiv.), C($sp^3$)–H coupling partner (1 mL), alkyl bromide (0.2 mmol, 1 equiv.) and stirred. Mixture 1 was then added to the reaction tube, capped with Teflon septum, and parafilmed. The reaction tube was removed from the glove box and irradiated using 34 W blue LEDs while stirring at RT (under fan cooling to keep the reaction at room temperature). After 48 h, the reaction was filtered through a small bed of celite and silica and concentrated *in vacuo*. The residue was purified by column chromatography using silica gel (100–200 mesh size) and DCM/hexane or Et$_2$O/pentane as the eluent. In the case of 1,4-dioxane and toluene coupling partners, two 34 W blue LEDs were used to irradiate for 96 h. All the compounds were fully characterized (see the Supplementary Information).

## Data availability

The authors declare that all other data supporting the findings of this study are available within the article and its Supplementary Information files. The experimental procedures and characterization of all new compounds are provided in Supplementary Information file. For the energies and Cartesian coordinates, see Supplementary Data 1 file.

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

## Acknowledgements

This publication is based upon work supported by the King Abdullah University of Science and Technology (KAUST), Office of Sponsored Research (OSR) under Award No. OSR-CRG2019-4025. The authors acknowledge the KAUST Supercomputing Laboratory for providing computational resources of the supercomputer Shaheen II.

## Author contributions

M.R., R.K., and K.M. conceived and designed the project. R.K., and K.M. performed and analyzed the experiments. S.K., R.K., and F.L. performed the spectroscopic studies. B.M., and L.C. did the computational studies. G.S.K. helped in preparing some of the starting materials. M.R., and R.K. wrote the manuscript with input from others. R.K., B.M., and S.K. prepared the supplementary information. M.R. directed the whole research.

## Competing interests

The authors declare no competing interests.
