## [Peer Review File · Nature Communications]

REVIEWER COMMENTS

Reviewer #1 (Remarks to the Author):

This manuscript describes a detailed study of the mechanism of nickel catalysed cross couplings enabled by visible light. DFT and experimental methods (nanosecond pump probe spectroscopy) are used to probe the elementary steps in this widely applied reaction and enable a more complete picture to be generated. This is a valuable aim; mechanistic and DFT studies of photochemically mediated processes are relatively rare and there are certainly challenges in bringing together the necessary methods to enable an insightful picture to be generated. In this paper, the authors have succeeded in doing that: the broad conclusions are (i) that energy transfer is responsible for promoting the nickel catalyst to an excited state and (ii) this liberates a bromine radical that performs H-atom abstraction and enables the cross coupling. This ultimately leads to a cross coupling between alkyl C-H substrates and sp^3 alkyl halides. This is a valuable reaction that enables the transformation of a commodity chemical (THF) into much higher value products. Although the yields and scope of this cross-coupling are modest in some cases, I believe this demonstrates the value of such a combined mechanistic-synthetic approach. I believe this paper will stimulate further development of reactions involving the triplet state of Ni(II).

Reviewer #2 (Remarks to the Author):

In this manuscript, Rueping and co-workers report the Nickel catalysed C(sp^3)-H alkylation of α -oxy and benzylic substrates with alkyl bromides, which is proposed to proceed via the photocatalytic sensitisation of nickel intermediates. This nickel catalysed C(sp^3)-H alkylation process is closely similar to the transformation reported by MacMillan and Koenig (cited), however it is proposed to proceed via a different mechanism due to variation in the reaction conditions (MacMillan uses an additional HAT catalyst, Koenig uses 4-CzIPN as photocatalyst). The study contains a thorough mechanistic investigation of the process, which supports an energy transfer pathway. However, a single electron transfer mechanism cannot be 100% ruled out. Whilst I have no doubt that the information delineated in this study will be useful for researchers in this field, I do not think that it is sufficiently impactful and of broad-interest for a leading journal such as Nature Communications. Regrettably, I am not supportive of publication. In addition, there are a significant numbers of points that need to be addressed.

Comments

- Page 2, line 11: "However, publications suggesting energy transfer (EnT) in metal-dual catalysis lack mechanistic understanding as EnT-pathways have not been experimentally explored."

There are a few points regarding this sentence. It inaccurately reflects the current state of the field and should be reworded.

1- change “metal-dual catalysis” to nickel catalysis – dual catalysis doesn’t make sense in this context.

2- there are papers which have carried out mechanistic studies/reactions on EnT in nickel catalysis, including the papers which the authors have cited 16-18, 19-21 – especially ref 21 which is a full mechanistic paper. Suggest rewording to “there is limited mechanistic understanding in nickel catalysis involving energy transfer.” or “there are only a few mechanistic studies which have been carried out on nickel catalysis involving energy transfer”.

- Page 4, line 3: “our initial test reaction between (3-bromopropyl)benzene and tetrahydrofuran (THF) using 2 mol% Ir[dF(CF₃)ppy]₂(dtbbpy)PF₆ (PS1), 5 mol% NiCl₂.glyme, 6 mol% 4,4’-di-tert-butyl-2,2’-bipyridine (4,4’-dtbbpy) and 2 equiv. of K₂CO₃ gave cross-coupled product 1 in 36% yield after visible light irradiation at RT. However, optimising various parameters, including changing the transition metal (TM) catalyst and ligand to (dOMe-*np*y)Ni(cod), boosted the yield to 77% (Table S1). Control experiments demonstrated that all the individual components e.g., [Ir], [Ni], base and visible light are necessary for the reaction to proceed.²⁹ Next, a number of photosensitizers, including strongly oxidizing Ru(bpz)₃.2PF₆ (PS3) and (9-MesAc)ClO₄ (PS4), were paired with nickel in a standard reaction to rule out the possibility of the reaction proceeding through a single electron transfer (SET) via oxidation of the Ni(II) complex to Ni(III), allowing the Ni(III) to catalyse the C-H functionalisation through a halogen photoelimination.³⁰⁻³³ However, these reactions did not yield any cross-coupled product, suggesting that a mechanism involving the oxidation of Ni(II) to Ni(III) might not be operative (Scheme 1C).”

There is a small scheme (scheme 1c, included with two introduction schemes 1a and 1b) which tries to summarise these results. This does not give enough information for the reader- in the scheme there are no reaction conditions, the exact substrate that this is carried out on is not given, no yield% (just says product was “formed”). It would be better for the reader to have a separate scheme on page 4 with a small optimisation table showing these results and the control reactions - instead of the results only contained in the text.

- My main criticism is that the reaction that is reported has not been mechanistically guided, despite the thrust of the paper being the mechanistic work. If this is not the case, then it doesn’t come across at all. I encourage the authors to put the scope directly after the reaction optimisation and before the mechanistic interrogation. The authors could then say then state that they will use this process as a tool to investigate the mechanism. Either way, there is little synthetic advance in terms of catalysis efficiency, despite the focus on the mechanistic work.

- Page 8, line 3: “Confirming this finding experimentally, (dtbbpy)Ni(II)-alkyl bromide prepared in situ displayed broad absorption features with λ_{max} values of 470 and 283 nm due to metal-to-ligand charge transfer (1MLCT) and a ligand-centered $\pi\text{-}\pi^*$ transition, respectively (Figure S1). Irradiating this in situ-generated complex with visible and ultraviolet light in the absence of PS1 either resulted in no-cross coupling product (with visible light) or trace product (with UV light), supporting the above conclusions”

Can the authors give more detail on what wavelengths were used for direct irradiation? It is claimed that the catalytic intermediate absorbs at 470 and 283, and so one might reasonably expect that light around 280nm would be beneficial if this is definitely an energy transfer mechanism. This seems like a rather obvious control experiment that should be included.

- Page 8, line 19: “Consistent with the experimental evidence, attempted cross-couplings in the presence of PS2, PS3 and PS4 (with singlet-triplet energy gaps of 50.5, 44.7 and 44.5 kcal/mol, respectively) showed no reactivity (Scheme 1 & Figure S18).

The authors originally used a different Ni ligand. A comment would be helpful. Can the energy gap for this intermediate be calculated? This would add further evidence for an energy transfer mechanism.

- Page 10, Fig 3 g) and Page 11, line 2: “At 2.6 mM of quencher Ds, bands at 470nm and 850 nm corresponding to 3^*PS1 got completely disappeared, and two new excited-state absorption (ESA) bands at 400 nm and 600 nm were observed along with a ground-state bleach (GSB) at 500 nm.”

The x-axis for this graph is from 400-800: whilst all other graphs are 400-1000. Why has this graph been cut short? Considering that the authors state that the band at 850 disappeared, the x-axis must be shown to 1000nm.

- Page 11, line 5: “This we tentatively assigned to triplet MLCT-state of nickel complex $^*\text{DT}$ resulting from triplet energy transfer from 3^*PS1 (Figure 3g). The observed ESA spectral features at 400 and 600 nm closely resemble the ESA spectral features of bpy radical anion, which supports our tentative assignment of MLCT state as the MLCT state contains a reduced bpy ligand.”

Can the authors provide an ns-TA spectra of the bpy radical anion? This statement is simply assumed without support. Does this prove energy transfer as the mechanism? Could it be postulated that a similar species could be formed through an electron transfer process from the photocatalyst?

- Page 11, line 15: "A separate catalytic reaction performed between THF and alkyl bromide using MK as photocatalyst which gave the sp³-sp³ cross-coupled product concluded that the EnT mechanism is operative (Scheme S1)."

It would be good to have this significant result as a scheme in the manuscript, otherwise this information could be glossed over.

- Page 13, table 1

Some of the yields in this scope are moderate to low: 43%, 47%, 39%, 24%, 26%, 31%. Considering that this is framed as a mechanism-guided paper, it is odd that the authors do not comment on this. Also, since the C-H coupling partner is used in massive excess as solvent, or 10 equiv when toluene is used, it seems that this is to suppress homocoupling of the alkyl halide. There is no mention of this except for a footnote in the table "c isolated in a 5:1 ratio with the homo-coupled product of alkyl halide". This really must be addressed.

- Page 14, line 3: "DFT and TD-DFT calculations point to a mechanism involving a Dexter triplet-triplet EnT from the *Ir(III) photosensitiser to the organometallic Ni(II) catalyst. The formation of the Ni(II) triplet state was further confirmed by nanosecond transient absorption spectroscopy, and the energy transfer rate constant was determined by time-resolved photoluminescence studies."

Whilst the calculations and the experiments point towards a sensitisation mechanism over electron transfer, it is not 100% conclusive. Therefore, avoid the phrase "confirmed".

Additional comments

- Abstract line 1: "Photoredox catalysis is an alternative to conventional synthetic processes for promoting organic reactions under mild conditions"

This sentence could be deleted.

- Abstract line 6: “sensitisation of organometallic catalysts”

This is misleading- it sounds like the authors can sensitise a range of organometallic species. Be specific “nickel catalytic intermediates”.

- Page 2, line 4-5: “reports of photocatalysis via energy transfer (EnT) are scarce”.

Reports of EnT are not scarce, there are many papers and reviews on this area. It is not as common as photoredox catalysis but scarce is very misleading.

- Page 2, line 13: “For example, the reductive elimination from ground state Ni(II) complexes was suggested in several publications, which however is not possible”.

This sentence is misleading. It sounds as if the mechanisms reported in the previous publications are completely incorrect. It is only that the previous papers suggested energy transfer was only required for the Ni-X bond homolysis, and not both the homolysis and reductive elimination. Rephrase to “Whilst energy transfer for the homolyses of Ni-X bond was invoked for the previous publications, reductive elimination from ground state Ni(II) complexes was suggested, which is....”

Reference 16-18 after “publications”, as I assume these are the ones that the authors are describing?

Reword “not possible” to unlikely.

- Page 5, line 1: “Thus, we report for the first time a full investigation of a photoredox and nickel catalysed C(sp³)-alkylation cross-coupling of ethers and alkyl bromides employing a combined experimental, computational and spectroscopic study”.

It’s not photoredox as the authors an suggest energy transfer mechanism.

- Page 5, line 9: “Next, a number of photosensitizers, including strongly oxidizing Ru(bpz)₃.2PF₆ (PS3) and (9-MesAcr)ClO₄ (PS4), were paired with nickel in a standard reaction to rule out the possibility of the reaction proceeding through a single electron transfer (SET) via oxidation of the Ni(II)”

In reference 18, a paper by the authors, (9-MesAcr)ClO₄ (PS4) is used and the authors propose an energy transfer mechanism for the nickel catalysed coupling of C(sp³)-H and aryl bromides. Why do they propose that this catalyst is fine for energy transfer in that case, but in this case it is used to rule out electron transfer? This should be addressed.

- Page 14, line 10: "Furthermore, the study highlights that for photoredox/metal-catalysed reactions next to SET pathways also energy transfer processes need to be considered"

A photoredox mechanism is a SET pathway.

Reviewer #3 (Remarks to the Author):

The paper by Rueping and co-workers is a very nice combined experimental / computational study of the mechanism in a very topical and original catalytic transformation. The reaction studied is C-C cross coupling involving nickel complexes in excited states generated by photosensitizers. The paper is very clearly written and the conclusions drawn are supported by adequate computational and photophysical studies. This contribution will be of interest to the readership of Nature Communications. I have only minor comments that the authors should address before publication :

1° The importance of the relative energy between the triplet state of the photosensitizer and the crucial reacting state of the triplet Ni complex is clearly described, explaining why certain photosensitizers are not effective. However, among the various possibilities from *PS1 to the various D(T) states (Figure 2), the one leading to reactivity is the least exoergic. Nevertheless the yield of the reaction is rather high. No discussion is provided about this apparent surprising observation. One would have expected the "wavefunction" of *PS1 to escape among the various triplet states for D(T), thus rendering the reaction not efficient. The authors should elaborate more on this aspect in the manuscript.

2° The work is presented in Scheme 1 and again in the conclusions as a new way of forming C(sp³)-C(sp³) bond for ether and benzylic positions. Yet there is only one example for benzylic position (toluene) and with a very low yield. I would consider it fairer not to mention functionalization of the benzylic position in this paper, unless more examples with increased yields are provided.

3° There is nobody among the authors listed bearing the affiliation to Aachen University. This typo should be corrected.

Reviewer #4 (Remarks to the Author):

The manuscript by Kancherla et al. deals with a comprehensive study on the Ni-catalyzed reaction between compounds with an activated C-H bond (tetrahydrofuran, 1,4-dioxane and toluene) and alkyl bromides. The reaction was found to occur upon visible light irradiation with an iridium complex (PS1) as the catalyst. Evidence was collected that the reaction proceeds by energy transfer from the photoexcited iridium complex to a Ni(II) intermediate, which is formed by oxidative addition into the C-Br bond of the alkyl bromides. Although I am not an expert in transient spectroscopy nor in DFT calculations, I consider the provided evidence convincing and I appreciate the combination of experimental synthetic work, spectroscopy, and theory. Given the interest in energy transfer as a possible vehicle to drive transition metal-catalyzed reactions I consider this contribution timely and valuable. It is recommended for publication after appropriate revision. The following aspects should be considered – in the order they appear in the manuscript:

(a) In the introduction, recent work by Bach on deracemization by energy transfer should be cited (Nature 2018, 564, 240; Angew. Chem. Int. Ed. 2020, 59, 21640).

(b) Sentence after the title “Results and Discussion”: “...we report a method for the direct...” (not “methodology”)

(c) Scheme 1: The energy refers to a mol of PS and the unit should be kcal/mol (actually it should be kJ/mol, but I have given up on asking people to use SI units...).

(d) The lifetime τ_0 for *PS1 is given first as 1.8 ± 0.003 microseconds which does not make sense because the precision of the first number is only one decimal. The same is true for the values given later (1.86 and 1.96). The confidence level should have the same decimal as the value it refers to.

(e) It should be “...mutual exchange of electrons...” instead of “...reverse transfer of electrons...”

(f) Despite that complex [E-F] may (Figure 2) provide a low energy transition state, it is a ternary complex and I do not envision this process to be more rapid than an initial hydrogen abstraction by the bromine radical and a subsequent combination of the tetrahydrofuryl radical with the Ni(I) complex E. This option should be at least mentioned.

(g) What do the authors mean with “Experimental details can be found ... elsewhere.”?

(h) Is there a reason why the quenching constant (Figure 3) was not calculated from the Stern-Volmer constant and the lifetime of *PS1. The purpose of plot 3c remains unclear.

(i) The discussion of Figure 3 should mention evidence for energy vs. electron transfer as obtained from the data of *PS1. Are the rate constants for bleach recovery and for the other transients

identical? As well documented (see for example the review by McCusker: Chem. Soc. Rev. 2016, 45, 5803), all TA traces should recover with the same rate if quenching occurs by energy transfer. A discussion why the mechanism is different from the mechanism of related reactions might be helpful (e.g. the work by Doyle, ref. 32).

(j) Unfortunately, there are no page or line numbers but the section after Figure 3 contains several minor errors: "...at 500 nm expected ..." (should read "...expected at 500 nm..."), "...got completely..." (should read "...completely..."), "...energy gaps..." (should read "...a...energy gap..."), "...concluded..." (should read "...confirmed..."), "...have shown..." (should read "...showed...", twice), "...have the excited..." (should read "...have an excited...").

(k) The KIE should also be determined from separate experiments but not only from competition experiments. As it stands, it remains unclear whether the C-H bond is cleaved in the turnover-limiting step or not (cf. ref. 48).

Response to the reviewers

We would like to thank the reviewers for their detailed evaluation and the constructive comments! We revised the manuscript and supporting information as per the reviewer's and editor's suggestions. We have highlighted the changes with "yellow background" in the revised manuscript and the details of our response for the respective comments are given below.

Reviewer #1 (Remarks to the Author):

This manuscript describes a detailed study of the mechanism of nickel catalysed cross couplings enabled by visible light. DFT and experimental methods (nanosecond pump probe spectroscopy) are used to probe the elementary steps in this widely applied reaction and enable a more complete picture to be generated. This is a valuable aim; mechanistic and DFT studies of photochemically mediated processes are relatively rare and there are certainly challenges in bringing together the necessary methods to enable an insightful picture to be generated. In this paper, the authors have succeeded in doing that: the broad conclusions are (i) that energy transfer is responsible for promoting the nickel catalyst to an excited state and (ii) this liberates a bromine radical that performs H-atom abstraction and enables the cross coupling. This ultimately leads to a cross coupling between alkyl C-H substrates and sp³ alkyl halides. This is a valuable reaction that enables the transformation of a commodity chemical (THF) into much higher value products. Although the yields and scope of this cross-coupling are modest in some cases, I believe this demonstrates the value of such a combined mechanistic-synthetic approach. I believe this paper will stimulate further development of reactions involving the triplet state of Ni(II).

We thank the reviewer for the positive and encouraging comment.

Reviewer #2 (Remarks to the Author):

In this manuscript, Rueping and co-workers report the Nickel catalysed C(sp³)-H alkylation of α -oxy and benzylic substrates with alkyl bromides, which is proposed to proceed via the photocatalytic sensitisation of nickel intermediates. This nickel catalysed C(sp³)-H alkylation process is closely similar to the transformation reported by MacMillan and Koenig (cited), however it is proposed to proceed via a different mechanism due to variation in the reaction conditions (MacMillan uses an additional HAT catalyst, Koenig uses 4-CzIPN as photocatalyst). The study contains a thorough mechanistic investigation of the process, which supports an energy transfer pathway. However, a single electron transfer mechanism cannot be 100% ruled out. Whilst I have no doubt that the information delineated in this study will be useful for researchers in this field, I do not think that it is sufficiently impactful and of broad-interest for a leading journal such as Nature Communications. Regrettably, I am not supportive of publication. In addition, there are a significant numbers of points that need to be addressed.

Comments

- Page 2, line 11: “However, publications suggesting energy transfer (EnT) in metal-dual catalysis

lack mechanistic understanding as EnT-pathways have not been experimentally explored.” There a few points regarding this sentence. It inaccurately reflects the current state of the field and should be reworded.

1- change “metal-dual catalysis” to nickel catalysis – dual catalysis doesn’t make sense in this context.

2- there are papers which have carried out mechanistic studies/reactions on EnT in nickel catalysis, including the papers which the authors have cited 16-18, 19-21 – especially ref 21 which is a full mechanistic paper. Suggest rewording to “there is limited mechanistic understanding in nickel catalysis involving energy transfer.” or “there are only a few mechanistic studies which have been carried out on nickel catalysis involving energy transfer”.

- Page 4, line 3: “our initial test reaction between (3-bromopropyl)benzene and tetrahydrofuran (THF) using 2 mol% Ir[dF(CF₃)ppy]₂(dtbbpy)PF₆ (PS1), 5 mol% NiCl₂.glyme, 6 mol% 4,4’-di-tert-butyl-2,2’-bipyridine (4,4’-dtbbpy) and 2 equiv. of K₂CO₃ gave cross-coupled product 1 in 36% yield after visible light irradiation at RT. However, optimising various parameters, including changing the transition metal (TM) catalyst and ligand to (dOMe-ppy)Ni(cod), boosted the yield to 77% (Table S1). Control experiments demonstrated that all the individual components e.g., [Ir], [Ni], base and visible light are necessary for the reaction to proceed.²⁹ Next, a number of photosensitizers, including strongly oxidizing Ru(bpz)₃.2PF₆ (PS3) and (9-MesAcr)ClO₄ (PS4), were paired with nickel in a standard reaction to rule out the possibility of the reaction proceeding through a single electron transfer (SET) via oxidation of the Ni(II) complex to Ni(III), allowing the Ni(III) to catalyse the C-H functionalisation through a halogen photoelimination.³⁰⁻³³ However, these reactions did not yield any cross-coupled product, suggesting that a mechanism involving the oxidation of Ni(II) to Ni(III) might not be operative (Scheme 1C).” There is a small scheme (scheme 1c, included with two introduction schemes 1a and 1b) which tries to summarise these results. This does not give enough information for the reader- in the scheme there are no reaction conditions, the exact substrate that this is carried out on is not given, no yield% (just says product was “formed”). It would be better for the reader to have a separate scheme on page 4 with a small optimisation table showing these results and the control reactions - instead of the results only contained in the text.

- My main criticism is that the reaction that is reported has not been mechanistically guided, despite the thrust of the paper being the mechanistic work. If this is not the case, then it doesn’t come across at all. I encourage the authors to put the scope directly after the reaction optimisation and before the mechanistic interrogation. The authors could then say then state that they will use this process as a tool to investigate the mechanism. Either way, there is little synthetic advance in terms of catalysis efficiency, despite the focus on the mechanistic work.

- Page 8, line 3: “Confirming this finding experimentally, (dtbbpy)Ni(II)-alkyl bromide prepared in situ displayed broad absorption features with λ_{max} values of 470 and 283 nm due to metal-to-ligand charge transfer (1MLCT) and a ligand-centered π - π^* transition, respectively (Figure S1). Irradiating this in situ-generated complex with visible and ultraviolet light in the absence of PS1 either resulted in no-cross coupling product (with visible light) or trace product (with UV light), supporting the above conclusions”

Can the authors give more detail on what wavelengths were used for direct irradiation? It is claimed that the catalytic intermediate absorbs at 470 and 283, and so one might reasonably expect that light around 280nm would be beneficial if this is definitely an energy transfer mechanism. This seems like a rather obvious control experiment that should be included.

- Page 8, line 19: “Consistent with the experimental evidence, attempted cross-couplings in the presence of PS2, PS3 and PS4 (with singlet-triplet energy gaps of 50.5, 44.7 and 44.5 kcal/mol, respectively) showed no reactivity (Scheme 1 & Figure S18). The authors originally used a different Ni ligand. A comment would be helpful. Can the energy gap for this intermediate be calculated? This would add further evidence for an energy transfer mechanism.

- Page 10, Fig 3 g) and Page 11, line 2: “At 2.6 mM of quencher Ds, bands at 470nm and 850 nm corresponding to 3^*PS1 got completely disappeared, and two new excited-state absorption (ESA) bands at 400 nm and 600 nm were observed along with a ground-state bleach (GSB) at 500 nm.” The x-axis for this graph is from 400-800: whilst all other graphs are 400-1000. Why has this graph been cut short? Considering that the authors state that the band at 850 disappeared, the x-axis must be shown to 1000nm.

- Page 11, line 5: “This we tentatively assigned to triplet MLCT-state of nickel complex *DT resulting from triplet energy transfer from 3^*PS1 (Figure 3g). The observed ESA spectral features at 400 and 600 nm closely resemble the ESA spectral features of bpy radical anion, which supports our tentative assignment of MLCT state as the MLCT state contains a reduced bpy ligand.”

Can the authors provide an ns-TA spectra of the bpy radical anion? This statement is simply assumed without support. Does this prove energy transfer as the mechanism? Could it be postulated that a similar species could be formed through an electron transfer process from the photocatalyst?

- Page 11, line 15: “A separate catalytic reaction performed between THF and alkyl bromide using MK as photocatalyst which gave the sp^3-sp^3 cross-coupled product concluded that the EnT mechanism is operative (Scheme S1).” It would be good to have this significant result as a scheme in the manuscript, otherwise this information could be glossed over.

- Page 13, table 1. Some of the yields in this scope are moderate to low: 43%, 47%, 39%, 24%, 26%, 31%. Considering that this is framed as a mechanism-guided paper, it is odd that the authors do not comment on this. Also, since the C-H coupling partner is used in massive excess as solvent, or 10 equiv when toluene is used, it seems that this is to suppress homocoupling of the alkyl halide. There is no mention of this except for a footnote in the table “c isolated in a 5:1 ratio with the homo-coupled product of alkyl halide”. This really must be addressed.

- Page 14, line 3: “DFT and TD-DFT calculations point to a mechanism involving a Dexter triplet-triplet EnT from the $^*Ir(III)$ photosensitiser to the organometallic $Ni(II)$ catalyst. The formation of the $Ni(II)$ triplet state was further confirmed by nanosecond transient absorption spectroscopy, and the energy transfer rate constant was determined by time-resolved photoluminescence studies.” Whilst the calculations and the experiments point towards a sensitisation mechanism over electron transfer, it is not 100% conclusive. Therefore, avoid the phrase “confirmed”. Additional comments

- Abstract line 1: “Photoredox catalysis is an alternative to conventional synthetic processes for promoting organic reactions under mild conditions” This sentence could be deleted.
- Abstract line 6: “sensitisation of organometallic catalysts” This is misleading- it sounds like the authors can sensitise a range of organometallic species. Be specific “nickel catalytic intermediates”.
- Page 2, line 4-5: “reports of photocatalysis via energy transfer (EnT) are scarce”. Reports of EnT are not scarce, there are many papers and reviews on this area. It is not as common as photoredox catalysis but scare is very misleading.
- Page 2, line 13: “For example, the reductive elimination from ground state Ni(II) complexes was suggested in several publications, which however is not possible”. This sentence is misleading. It sounds as if the mechanisms reported in the previous publications are completely incorrect. It is only that the previous papers suggested energy transfer was only required for the Ni-X bond homolysis, and not both the homolysis and reductive elimination. Rephrase to “Whilst energy transfer for the homolyses of Ni-X bond was invoked for the previous publications, reductive elimination from ground state Ni(II) complexes was suggested, which is...” Reference 16-18 after “publications”, as I assume these are the ones that the authors are describing? Reword “not possible” to unlikely.
- Page 5, line 1: “Thus, we report for the first time a full investigation of a photoredox and nickel catalysed C(sp³)-alkylation cross-coupling of ethers and alkyl bromides employing a combined experimental, computational and spectroscopic study”. It’s not photoredox as the authors an suggest energy transfer mechanism.
- Page 5, line 9: “Next, a number of photosensitizers, including strongly oxidizing Ru(bpz)₃.2PF₆ (PS3) and (9-MesAcr)ClO₄ (PS4), were paired with nickel in a standard reaction to rule out the possibility of the reaction proceeding through a single electron transfer (SET) via oxidation of the Ni(II)”
In reference 18, a paper by the authors, (9-MesAcr)ClO₄ (PS4) is used and the authors propose an energy transfer mechanism for the nickel catalysed coupling of C(sp³)-H and aryl bromides. Why do they propose that this catalyst is fine for energy transfer in that case, but in this case it is used to rule out electron transfer? This is should be addressed.
- Page 14, line 10: “Furthermore, the study highlights that for photoredox/metal-catalysed reactions next to SET pathways also energy transfer processes need to be considered”. A photoredox mechanism is a SET pathway.

We thank the reviewer the evaluation of our manuscript and for constructive suggestions which we included in the revised manuscript.

We believe our work is impactful and of broad interest since it involves experimental, computational, and transient spectroscopic techniques and provides new insights.

Before addressing the individual comments, we would like to provide a brief discussion and arguments for an energy transfer mechanism and against a SET pathway.

The explanation for ruling out the electron transfer (ET) mechanism: If the reaction proceeds through the ET pathway via Ni(III) intermediate, an electron from quencher \mathbf{D}^{S} (i.e. Ni(II)) enters into an empty t_{2g} orbital of excited state $^3\mathbf{PS1}$ (i.e. Ir) which leads to the formation of anionic Ir(II) and cationic Ni(III) species. This step is exergonic by 21.4 kcal/mol (Supplementary Fig. 15). However, the high energy barrier for the following C–H activation step ($\Delta G^\ddagger = 45.9$ kcal/mol) prevents further progress along this pathway confirming that it is not a product-forming channel.

A similar conclusion also can be obtained from the transient spectroscopic studies. The excited state $^3\mathbf{PS1}$ contains oxidized Ir (one electron empty t_{2g}) and reduced ligand (occupied π^*). In Fig. 3d, ns-TA spectra of $^3\mathbf{PS1}$ (0.1 mM) in the absence of quencher \mathbf{D}^{S} display two broad excited state absorption (ESA) peaks at 470 and 850 nm which corresponds to the reduced bipyridine ligands in the Ir photocatalyst. Similarly, oxidized Ir should appear as a ground state bleach (GSB) that falls below 400 nm, which we cannot see due to instrument limitation. Assuming the reaction is proceeding via a Ni(III) intermediate, then by adding the quencher \mathbf{D}^{S} (Ni(II)-alkyl bromide) to the excited state Ir photocatalyst, an electron from Ni(II) enters into the empty t_{2g} orbital of excited state Ir. Therefore, with the increase in the concentration of the Ni(II) quencher, the ESA peaks (470 and 850 nm) corresponding to the reduced bipyridine ligand of photocatalyst should not be affected. Simultaneously, new TA signals corresponding to Ni(III) should appear. Overall, the quenching rate for these TA signals should be different if it proceeds via the ET pathway as Reviewer-4 highlighted (*Chem. Soc. Rev.* **2016**, *45*, 5803-5820). However, Fig. 3d-g clearly shows that ESA peaks at 470 and 850 nm have disappeared with the increase in the concentration of quencher, and new transient peaks (both ESA and GSB) have appeared. As discussed above, the ESA peaks corresponding to the reduced ligand (occupied π^*) of the Ir photocatalyst should not be affected if the reaction proceeds via the Ir(II)/Ni(III) pathway. However, this is not occurring in our system, which shows that an alternate energy transfer pathway is operative. Moreover, the quenching rate (450-480, 775-880 nm) and the formation rate (590-650 nm) of the transient signals are identical (Fig. 4a) which supports our conclusion.

Explanation supporting the energy transfer (EnT) mechanism: If the reaction proceeds through the EnT pathway, then one would expect the mutual exchange of electrons between excited $^3\mathbf{PS1}$ and Ni(II)-alkyl bromide (Fig. 1, and Supplementary Fig. 17). To realize the product formation, an electron from the occupied π^* orbital in the $^3\mathbf{PS1}$ photocatalyst should be promoted to the Ni–Br σ^* orbital (Supplementary Fig. 16, it is involved in the \mathbf{T}^5 excited state, $^*\mathbf{D}^{\text{T}5}$, with a singlet-triplet energy gap, $\Delta G_{\text{T}5-\text{S}0}$, of 56.2 kcal/mol). Simultaneously an electron from the ground state Ni(II) will be promoted to empty t_{2g} orbital of $^3\mathbf{PS1}$ gives rise to excited Ni(II) and ground-state Ir(III). When photocatalysts PS1-PS4 with different triplet energies and redox potentials were evaluated, only Ir[dF(CF₃)ppy]₂(dtbbpy)PF₆ (**PS1**) with lowest singlet-triplet energy gap ($\Delta G_{\text{T}1-\text{S}0} = 60.7$ kcal/mol) larger than five singlet-triplet energy gaps for Ni(II)-alkyl bromide \mathbf{D}^{S} (Supplementary Fig. 18) gave the product. This shows that the reaction proceeds via the EnT pathway but not by the ET mechanism.

Spectroscopically, mutual exchange of electrons between excited $^3\text{PS1}$ and Ni(II)-alkyl bromide should result in the decay of ESA transient signal at 470 and 850 nm (corresponds to the reduced bipyridine ligand in the Ir photocatalyst), since it transfers that electron to the T^5 excited state of Ni(II). Simultaneously an electron from the ground state Ni(II) will be promoted to empty t_{2g} orbital of $^3\text{PS1}$ giving rise to ground-state Ir(III) and excited Ni(II). Therefore the overall transient spectral changes involve the decay of ESA (at 470 and 850 nm), and the appearance of new transient signals corresponds to excited Ni(II). Also, the excited Ni(II) should display a long lifetime as the mutual exchange of electrons results in the formation of spin-flipped Ni(II) triplet state. As expected, the excited $^3\text{PS1}$ lifetime monotonically decreased upon the addition of the quencher D^{S} and a new ESA band has appeared at 600 nm (Fig. 3e). In order to have better visibility of the EnT product forming step, an excessive amount of quencher Ni(II) was added. At 2.6 mM of quencher D^{S} , ESA bands at 470 nm and 850 nm corresponding to $^3\text{PS1}$ disappeared, and two new ESA bands at 400 and 600 nm were observed along with a GSB at 500 nm. The TA bands at 450-480 and 775-880 nm were found to decay at the same rate with the generation of the new band formed at 590-650 nm (Fig. 4a), thus supporting that the quenching is by the EnT mechanism. The newly formed transient species displayed a long-lived excited-state lifetime of $\tau = 671$ ns and we assign this newly formed long-lived excited state of Ni to be a spin-flipped triplet state. Further confirmation for the EnT mechanism was also obtained by performing a separate ns-TA on a mixture of Michler's ketone (singlet-triplet energy gaps of 61.0 kcal/mol) and Ni(II)-alkyl bromide which also displayed the same transient features at 400 and 600 nm (Fig. 3h). We initially assigned the new state formed as an MLCT state of nickel (based on the work by Doyle and co-workers, *J. Am. Chem. Soc.* **2018**, *140*, 3035–3039). However, we later realized that making such an assignment is inappropriate since our newly formed spin-flipped Ni-triplet state has a very long lifetime compared to that of the triplet state of nickel shown by Doyle and Macmillan groups (*J. Am. Chem. Soc.* **2020**, *142*, 5800–5810; *J. Am. Chem. Soc.* **2020**, *142*, 4555–4559).

Reviewer's Comment 1: Page 2, line 11: a) change “metal-dual catalysis” to nickel catalysis – dual catalysis doesn't make sense in this context. b) there are papers that have carried out mechanistic studies/reactions on EnT in nickel catalysis, including the papers which the authors have cited 16-18, 19-21 – especially ref 21 which is a full mechanistic paper. Suggest rewording to “there is limited mechanistic understanding in nickel catalysis involving energy transfer.” or “there are only a few mechanistic studies which have been carried out on nickel catalysis involving energy transfer”.

Answer: References 16-20 are experimental papers, and reference 21 is purely mechanistic paper. Still, due to the intrinsic limitation, new transient peaks corresponding to EnT cannot be seen even in reference 21.

Following the reviewer's suggestions, we changed the sentence to “However, there is limited mechanistic understanding in nickel catalysis involving energy transfer.”

Reviewer's Comment 2: Page 4, There is a small scheme (scheme 1c, included with two introduction schemes 1a and 1b) that tries to summarise these results. This does not give enough information for the reader- in the scheme, there are no reaction conditions, the exact substrate that this is carried out on is not given, no yield% (just says the product was “formed”). It would be better for the reader to have a separate scheme on page 4 with a small optimisation table showing these results and the control reactions - instead of the results only contained in the text.

Answer: Following the reviewer's suggestion, to help the readers understand, a new optimization Table 1 is included on page 5 along with the running text.

Reviewer's Comment 3: My main criticism is that the reaction that is reported has not been mechanistically guided, despite the thrust of the paper being the mechanistic work. If this is not the case, then it doesn't come across at all. I encourage the authors to put the scope directly after the reaction optimisation and before the mechanistic interrogation. The authors could then say then state that they will use this process as a tool to investigate the mechanism. Either way, there is little synthetic advance in terms of catalysis efficiency, despite the focus on the mechanistic work.

Answer: Following the reviewer's suggestion, the scope of substrates is now placed on page 6 after the optimization table.

Reviewer's Comment 4: Page 8, line 3: “Confirming this finding experimentally, (dtbbpy)Ni(II)-alkyl bromide prepared in situ displayed broad absorption features with λ_{max} values of 470 and 283 nm due to metal-to-ligand charge transfer (1MLCT) and a ligand-centered π - π^* transition, respectively (Figure S1). Irradiating this in situ-generated complex with visible and ultraviolet light in the absence of PS1 either resulted in no-cross coupling product (with visible light) or trace product (with UV light), supporting the above conclusions.”

Can the authors give more detail on what wavelengths were used for direct irradiation? It is claimed that the catalytic intermediate absorbs at 470 and 283, and so one might reasonably expect that light around 280nm would be beneficial if this is definitely an energy transfer mechanism. This seems like a rather obvious control experiment that should be included.

Answer: We used a Rayonet photochemical reactor using a 300 nm wavelength of light. The logic behind this reaction is that we should not expect product formation without using an external photocatalyst, since both the absorptions (470 and 283 nm) from the Ni(II)-alkyl bromide intermediate **D^S** corresponds to singlet metal-to-ligand charge transfer (¹MLCT) and a ligand-centered π - π^* transitions respectively.

To realize the cross-coupling, one electron from an occupied molecular orbital should be promoted to the Ni–Br σ^* orbital, which triggers the breaking of the Ni–Br bond. Analysis of the orbitals contributing to the dominant electronic transitions of **D^S** in the UV-Vis region revealed that all the excitations correspond to transitions of electrons from occupied Ni orbitals to the LUMO, LUMO+1, and LUMO+2, which represent π^* molecular orbitals located on the ligand (Supplementary Table 2, and Supplementary Fig. 16). These excitations are not effective in promoting the reactivity of the Ni(II) complex since the electron is not promoted to the Ni–Br σ^*

orbital (LUMO+3). Analysis of the molecular orbitals of \mathbf{D}^S indicates that LUMO+3, which essentially corresponds to the Ni–Br σ^* orbital (Supplementary Fig. 16), is involved in the \mathbf{T}^5 excited state, $^*\mathbf{D}^{\mathbf{T}5}$, with a singlet-triplet energy gap, $\Delta G_{\mathbf{T}5-S0}$, of 56.2 kcal/mol. Therefore, to realize the product formation an external photocatalyst with energy greater than 56.2 kcal/mol is necessary.

The wavelength of UV light used is now given in the revised manuscript (page 10).

Reviewer's Comment 5: Page 8, line 19: “Consistent with the experimental evidence, attempted cross-couplings in the presence of PS2, PS3 and PS4 (with singlet-triplet energy gaps of 50.5, 44.7 and 44.5 kcal/mol, respectively) showed no reactivity (Scheme 1 & Figure S18).

The authors originally used a different Ni ligand. A comment would be helpful. Can the energy gap for this intermediate be calculated? This would add further evidence for an energy transfer mechanism.

Answer: We would like to clarify that all the control experiments were carried out using the optimized reaction condition. In all these reactions, we have used a 4,4'-dOMe-bpy ligand which is also used for DFT calculations purposes. This confusion might have aroused since we did not give the optimization table in the manuscript. We now included the optimization table in the revised manuscript.

Reviewer's Comment 6: Page 10, Fig 3 g, and Page 11, line 2: “At 2.6 mM of quencher Ds, bands at 470nm and 850 nm corresponding to 3^*PS1 got completely disappeared, and two new excited-state absorption (ESA) bands at 400 nm and 600 nm were observed along with a ground-state bleach (GSB) at 500 nm.”

The x-axis for this graph is from 400-800: whilst all other graphs are 400-1000. Why has this graph been cut short? Considering that the authors state that the band at 850 disappeared, the x-axis must have been shown to 1000nm.

Answer: One reason why the graph has been cut short is that the signal has already reached zero by 800 nm wavelength. It is clear from Fig. 3d-f, the peak at 850 nm we described is very broad ranging from 700-1000 nm, with absorption maxima at 850 nm. However, in Fig. 3g, such a broad feature from 700-1000 nm, with absorption maxima at 850 nm completely disappeared and it reached zero at 800 nm itself.

The second reason is, for the longer time scale the signal after 800 nm is very noisy and the signal-to-noise ratio is poor (as shown in the picture). Therefore, to have better clarity, we have cut short the graph to 400-800 nm.

Reviewer's Comment 7: Page 11, line 5: “This we tentatively assigned to triplet MLCT-state of nickel complex $^*\mathbf{DT}$ resulting from triplet energy transfer from 3^*PS1 (Figure 3g). The observed ESA spectral features at 400 and 600 nm closely resemble the ESA spectral features of bpy radical

anion, which supports our tentative assignment of MLCT state as the MLCT state contains a reduced bpy ligand.”

Can the authors provide an ns-TA spectra of the bpy radical anion? This statement is simply assumed without support. Does this prove energy transfer as the mechanism? Could it be postulated that a similar species could be formed through an electron transfer process from the photocatalyst?

Answer: The detailed explanation for ruling out the ET pathway thus supporting the EnT mechanism is discussed (*vide supra*). As discussed above, we revise our new long-lived excited state as a spin-flipped Ni-triplet state formed by EnT from the photosensitizer.

The exact nature of the Ni-triplet state formed cannot be assigned due to the limited reports on the nature of transient signals formed via EnT in nickel catalysis. A recent publication by MacMillan and co-workers also could not assign the exact nature of the Ni-triplet state due to the intrinsic limitation (*J. Am. Chem. Soc.* **2020**, *142*, 4555–4559).

Reviewer’s Comment 8: Page 11, line 15: “A separate catalytic reaction performed between THF and alkyl bromide using MK as photocatalyst which gave the sp³-sp³ cross-coupled product concluded that the EnT mechanism is operative (Scheme S1).”

It would be good to have this significant result as a scheme in the manuscript, otherwise this information could be glossed over.

Answer: As suggested by the reviewer, the reaction with MK is given in the revised manuscript as a separate scheme (Fig. 4c).

Reviewer’s Comment 9: Page 13, table 1, Some of the yields in this scope are moderate to low: 43%, 47%, 39%, 24%, 26%, 31%. Considering that this is framed as a mechanism-guided paper, it is odd that the authors do not comment on this. Also, since the C-H coupling partner is used in massive excess as solvent, or 10 equiv when toluene is used, it seems that this is to suppress homocoupling of the alkyl halide. There is no mention of this except for a footnote in the table “c isolated in a 5:1 ratio with the homo-coupled product of alkyl halide”. This really must be addressed.

Answer: In the case of 1,4-dioxane, the diminished reactivity (24% and 26%) might be due to polar effects and the fact that the second oxygen atom, located in the β -position with respect to the reaction site, disfavors hydrogen atom abstraction by an electrophilic radical.^{38,39} Also, inductive effects might be the reason for diminished reactivity.¹⁸ In case of compound **17** steric effect play a major role due to the bulkiness of the alkyl bromide, which is resulting in the formation of hydrogenation side product rather than cross-coupled product. In case of compound **5** we believe that the Ni(II) intermediate is undergoing β -hydride elimination resulting in the formation of olefinated side product thus decreasing the yield. In case of toluene **28** diminished reactivity might be due to the exceptional stability of the benzylic radical.

We tried to use THF in equivalent amounts rather than using it as a solvent, however, it resulted in homocoupling of alkyl halide as a major product. Therefore, we used THF as solvent amount to suppress the homocoupling and to promote the cross-coupling reaction.

The related discussion is included in the revised manuscript on page 8.

Reviewer's Comment 10: Page 14, line 3: "DFT and TD-DFT calculations point to a mechanism involving a Dexter triplet-triplet EnT from the *Ir(III) photosensitizer to the organometallic Ni(II) catalyst. The formation of the Ni(II) triplet state was further confirmed by nanosecond transient absorption spectroscopy, and the energy transfer rate constant was determined by time-resolved photoluminescence studies." Whilst the calculations and the experiments point towards a sensitisation mechanism over electron transfer, it is not 100% conclusive. Therefore, avoid the phrase "confirmed".

Answer: The detailed explanation for ruling out the ET pathway thus supporting the EnT pathway is discussed (*vide supra*).

However, the term "confirmed" is changed to "supported".

Additional comments

Reviewer's Comment 11: Abstract line 1: "Photoredox catalysis is an alternative to conventional synthetic processes for promoting organic reactions under mild conditions" - This sentence could be deleted.

Answer: Following the reviewer's suggestion, this sentence is deleted in the revised manuscript.

Reviewer's Comment 12: Abstract line 6: "sensitisation of organometallic catalysts" - This is misleading- it sounds like the authors can sensitise a range of organometallic species. Be specific "nickel catalytic intermediates".

Answer: Following the reviewer's suggestion, sensitisation of organometallic catalysts is changed to "sensitisation of nickel catalytic intermediates".

Reviewer's Comment 13: Page 2, line 4-5: "reports of photocatalysis via energy transfer (EnT) are scarce".

Reports of EnT are not scarce, there are many papers and reviews on this area. It is not as common as photoredox catalysis but scarce is very misleading.

Answer: The term scarce is changed to "limited".

Reviewer's Comment 14: Page 2, line 13: "For example, the reductive elimination from ground state Ni(II) complexes was suggested in several publications, which however is not possible".

This sentence is misleading. It sounds as if the mechanisms reported in the previous publications are completely incorrect. It is only that the previous papers suggested energy transfer was only required for the Ni-X bond homolysis, and not both the homolysis and reductive elimination. Rephrase to "Whilst energy transfer for the homolyses of Ni-X bond was invoked for

the previous publications, reductive elimination from ground state Ni(II) complexes was suggested, which is....” Reference 16-18 after “publications”, as I assume these are the ones that the authors are describing? Reword “not possible” to unlikely.

Answer: Following the reviewer's suggestion, the sentence is now changed to “For example, whilst energy transfer for the homolysis of Ni-halogen bond was invoked for the previous publications,¹⁸⁻²⁰ reductive elimination from ground state Ni(II) complexes was suggested, which is unlikely.”

Reviewer's Comment 15: Page 5, line 1: “Thus, we report for the first time a full investigation of a photoredox and nickel catalysed C(sp³)-alkylation cross-coupling of ethers and alkyl bromides employing a combined experimental, computational and spectroscopic study”. It's not photoredox as the authors and suggest energy transfer mechanism.

Answer: The term photoredox is changed to “photochemical nickel catalyzed.”

Reviewer's Comment 16: Page 5, line 9: “Next, a number of photosensitizers, including strongly oxidizing Ru(bpz)₃.2PF₆ (PS3) and (9-MesAcr)ClO₄ (PS4), were paired with nickel in a standard reaction to rule out the possibility of the reaction proceeding through a single electron transfer (SET) via oxidation of the Ni(II)”.

In reference 18, a paper by the authors, (9-MesAcr)ClO₄ (PS4) is used and the authors propose an energy transfer mechanism for the nickel catalysed coupling of C(sp³)-H and aryl bromides. Why do they propose that this catalyst is fine for energy transfer in that case, but in this case it is used to rule out electron transfer? This is should be addressed.

Answer: The choice of the photocatalyst for any ET or EnT process is completely dependent on their redox potentials and triplet energies. In the reported arylation reaction (reference 18) the corresponding C(sp³)-H activation occurs at the excited state of LNi(II)(Ar)Br. The energy levels of excited states of LNi(II)(Ar)Br might be different than the corresponding LNi(II)(alkyl)Br in the current alkylation paper. Therefore, due to the different energy levels of LNi(II)(Ar)Br, the photocatalyst (9-MesAcr)ClO₄ (PS4) might be suitable. However, in-depth analysis of excited-state energy levels and electronic structures of LNi(II)(Ar)Br is underway in a different project.

Reviewer's Comment 17: Page 14, line 10: “Furthermore, the study highlights that for photoredox/metal-catalysed reactions next to SET pathways also energy transfer processes need to be considered” - A photoredox mechanism is a SET pathway.

Answer: The term photoredox/metal-catalysed reactions is changed to “photochemical/metal-catalyzed reactions”

Reviewer #3 (Remarks to the Author):

The paper by Rueping and co-workers is a very nice combined experimental / computational study of the mechanism in a very topical and original catalytic transformation. The reaction studied is C-C cross coupling involving nickel complexes in excited states generated by photosensitizers. The paper is very clearly written and the conclusions drawn are supported by adequate computational and photophysical studies. This contribution will be of interest to the readership of Nature Communications. I have only minor comments that the authors should address before publication:

1° The importance of the relative energy between the triplet state of the photosensitizer and the crucial reacting state of the triplet Ni complex is clearly described, explaining why certain photosensitizers are not effective. However, among the various possibilities from *PS1 to the various D(T) states (Figure 2), the one leading to reactivity is the least exoergic. Nevertheless the yield of the reaction is rather high. No discussion is provided about this apparent surprising observation. One would have expected the "wavefunction" of *PS1 to escape among the various triplet states for D(T), thus rendering the reaction not efficient. The authors should elaborate more on this aspect in the manuscript.

2° The work is presented in Scheme 1 and again in the conclusions as a new way of forming C(sp³)-C(sp³) bond for ether and benzylic positions. Yet there is only one example for benzylic position (toluene) and with a very low yield. I would consider it fairer not to mention functionalization of the benzylic position in this paper, unless more examples with increased yields are provided

3° There is nobody among the authors listed bearing the affiliation to Aachen University. This typo should be corrected

We thank the reviewer for the positive and constructive comments.

Reviewer's Comment 1: The importance of the relative energy between the triplet state of the photosensitizer and the crucial reacting state of the triplet Ni complex is clearly described, explaining why certain photosensitizers are not effective. However, among the various possibilities from *PS1 to the various D(T) states (Figure 2), the one leading to reactivity is the least exoergic. Nevertheless, the yield of the reaction is rather high. No discussion is provided about this apparent surprising observation. One would have expected the "wave function" of *PS1 to escape among the various triplet states for D(T), thus rendering the reaction not efficient. The authors should elaborate more on this aspect in the manuscript.

Answer: We thank the reviewer for the comment, however, we do not completely agree with the reviewer in this regard. In the energy transfer state, the excitation of $D^S \rightarrow *D^{T5}$ requires 56.2 kcal/mol energy, while the corresponding deactivation of $*PS1 \rightarrow PS1$ loses 60.7 kcal/mol. Therefore, the overall energy transfer step is exergonic by 4.5 kcal/mol. The next step is C-H bond activation, which starts from $*D^{T5}$ state and thus the free energy barrier should be counted from that state. Therefore, the overall activation barrier for the C-H activation step is 9.4 kcal/mol, which is quite low explaining the high product yield.

Reviewer's Comment 2: The work is presented in Scheme 1 and again in the conclusions as a new way of forming C(sp³)-C(sp³) bond for ether and benzylic positions. Yet there is only one example for benzylic position (toluene) and with a very low yield. I would consider it fairer not to mention functionalization of the benzylic position in this paper, unless more examples with increased yields are provided.

Answer: Following the reviewer's suggestion, functionalization of the benzylic position is not mentioned in the revised manuscript.

Reviewer's Comment 3: There is nobody among the authors listed bearing the affiliation to Aachen University. This typo should be corrected.

Answer: Typo corrected.

Reviewer #4 (Remarks to the Author): The manuscript by Kancherla et al. deals with a comprehensive study on the Ni-catalyzed reaction between compounds with an activated C-H bond (tetrahydrofuran, 1,4-dioxane and toluene) and alkyl bromides. The reaction was found to occur upon visible light irradiation with an iridium complex (PS1) as the catalyst. Evidence was collected that the reaction proceeds by energy transfer from the photoexcited iridium complex to a Ni(II) intermediate, which is formed by oxidative addition into the C-Br bond of the alkyl bromides. Although I am not an expert in transient spectroscopy nor in DFT calculations, I consider the provided evidence convincing and I appreciate the combination of experimental synthetic work, spectroscopy, and theory. Given the interest in energy transfer as a possible vehicle to drive transition metal-catalyzed reactions I consider this contribution timely and valuable. It is recommended for publication after appropriate revision. The following aspects should be considered – in the order they appear in the manuscript:

(a) In the introduction, recent work by Bach on deracemization by energy transfer should be cited (Nature 2018, 564, 240; Angew. Chem. Int. Ed. 2020, 59, 21640).

(b) Sentence after the title “Results and Discussion”: “...we report a method for the direct...” (not “methodology”)

(c) Scheme 1: The energy refers to a mol of PS and the unit should be kcal/mol (actually it should be kJ/mol, but I have given up on asking people to use SI units...).

(d) The lifetime τ_0 for *PS1 is given first as 1.8 ± 0.003 microseconds which does not make sense because the precision of the first number is only one decimal. The same is true for the values given later (1.86 and 1.96). The confidence level should have the same decimal as the value it refers to.

(e) It should be “...mutual exchange of electrons...” instead of “...reverse transfer of electrons...”

(f) Despite that complex [E-F] may (Figure 2) provide a low energy transition state, it is a ternary complex and I do not envision this process to be more rapid than an initial hydrogen abstraction by the bromine radical and a subsequent combination of the tetrahydrofuryl radical with the Ni(I) complex E. This option should be at least mentioned.

(g) What do the authors mean with “Experimental details can be found ... elsewhere.”?

(h) Is there a reason why the quenching constant (Figure 3) was not calculated from the Stern-Volmer constant and the lifetime of *PS1. The purpose of plot 3c remains unclear.

(i) The discussion of Figure 3 should mention evidence for energy vs. electron transfer as obtained from the data of *PS1. Are the rate constants for bleach recovery and for the other transients identical? As well documented (see for example the review by McCusker: Chem. Soc. Rev. 2016, 45, 5803), all TA traces should recover with the same rate if quenching occurs by energy transfer. A discussion why the mechanism is different from the mechanism of related reactions might be helpful (e.g. the work by Doyle, ref. 32).

(j) Unfortunately, there are no page or line numbers but the section after Figure 3 contains several minor errors: “...at 500 nm expected ...” (should read “...expected at 500 nm...”), “...got completely...” (should read “...completely...”), “...energy gaps...” (should read “...a...energy

gap...”), “...concluded...” (should read “...confirmed...”), “...have shown...” (should read “...showed...”, twice), “...have the excited...” (should read “...have an excited...”).

(k) The KIE should also be determined from separate experiments but not only from competition experiments. As it stands, it remains unclear whether the C-H bond is cleaved in the turnover-limiting step or not (cf. ref. 48).

We thank the reviewer for the positive and constructive comments.

Reviewer’s Comment 1: In the introduction, recent work by Bach on deracemization by energy transfer should be cited (Nature 2018, 564, 240; Angew. Chem. Int. Ed. 2020, 59, 21640).

Answer: Following the reviewer’s suggestion, recent work by Bach is cited (reference 13 and 14).

Reviewer’s Comment 2: Sentence after the title “Results and Discussion”: “...we report a method for the direct...” (not “methodology”)

Answer: Following the reviewer’s suggestion, methodology is changed to “method”

Reviewer’s Comment 3: Scheme 1: The energy refers to a mol of PS and the unit should be kcal/mol (actually it should be kJ/mol, but I have given up on asking people to use SI units...).

Answer: Following the reviewer’s suggestion, kcal is corrected to “kcal/mol” in scheme 1.

Reviewer’s Comment 4: The lifetime τ_0 for *PS1 is given first as 1.8 ± 0.003 microseconds which does not make sense because the precision of the first number is only one decimal. The same is true for the values given later (1.86 and 1.96). The confidence level should have the same decimal as the value it refers to.

Answer: Following the reviewer’s suggestion, the precision value is corrected to $\tau_0 = 1.865 \pm 0.003$ μs and 1.967 ± 0.002 μs .

Reviewer’s Comment 5: It should be “...mutual exchange of electrons...” instead of “...reverse transfer of electrons...”

Answer: Following the reviewer’s suggestion, “reverse transfer of electron” is changed to “mutual exchange of electrons”

Reviewer’s Comment 6: Despite that complex [E-F] may (Figure 2) provide a low energy transition state, it is a ternary complex and I do not envision this process to be more rapid than an initial hydrogen abstraction by the bromine radical and a subsequent combination of the tetrahydrofuryl radical with the Ni(I) complex E. This option should be at least mentioned.

Answer: We are thankful to the reviewer for this valuable comment. According to the reviewer’s suggestion, we have calculated an alternative pathway of C α -H bond activation by the active radical. Our calculations suggest that this alternative pathway is unfavorable. The following Figure and text are added in the Supplementary Information.

Supplementary Fig. 20. a) PES (potential energy surface) and b) free energy of C_α -H bond activation of THF by active Br atom.

In an alternative pathway, the outer-sphere mechanism of C_α -H activation by active Br atom has been considered. In the PES (potential energy surface) energy is gradually increasing along the coordinate of H and Br bond formation (Supplementary Fig. 20). We have selected one structure for single-point energy calculation from the PES where the H-Br bond is already formed. That structure is unstable by 2.3 kcal/mol than the inner sphere transition state $[E-F]^\ddagger$. Finally, the outer sphere mechanism by the active Br atom is less favored than the inner-sphere mechanism.

Reviewer's Comment 7: What do the authors mean with "Experimental details can be found ... elsewhere."?

Answer: We mean that the instrument details can be found in our previous publications. We now changed it to "Experimental details can be found in the SI and from our previous publications.^{47,48}" with suitable citations.

Reviewer's Comment 8: Is there a reason why the quenching constant (Figure 3) was not calculated from the Stern-Volmer constant and the lifetime of 3PS1 . The purpose of plot 3c remains unclear.

Answer: The quenching constant $k_q = 8.16 \times 10^9 \text{ M}^{-1} \text{ s}^{-1}$ is now given in the revised manuscript. Plot 3c was used to determine Triplet-Triplet energy transfer rate constant (TTEnT) and also to see if there is any possibility of the reverse TTEnT. The following discussion is included in the revised manuscript. "Furthermore, the triplet-triplet EnT rate constant k_{TTEnT} was determined by TRPL measurements. A k_{TTEnT} of $(7.95 \pm 0.31) \times 10^9 \text{ L mol}^{-1} \text{ s}^{-1}$ was determined from the linear fit of the plot of the observed (measured) rate constant (k_{obs}) corrected by the ground-state recovery rate (k_{GSR}) of 3PS1 (obtained in the absence of D^S) versus different concentrations of D^S (Fig. 3c, also see Supplementary Fig. 7). Since there is no sizeable intercept, we believe that the reverse TTEnT might not be operative.^{17,43,44}"

Reviewer's Comment 9: The discussion of Figure 3 should mention evidence for energy vs. electron transfer as obtained from the data of *PS1. Are the rate constants for bleach recovery and for the other transients identical? As well documented (see for example the review by McCusker: Chem. Soc. Rev. 2016, 45, 5803), all TA traces should recover with the same rate if quenching occurs by energy transfer.

Answer: The detailed explanation for ruling out the ET pathway thus supporting the EnT pathway is discussed (*vide supra*).

The TA bands at 450-480 and 775-880 nm were found to decay at the same rate with the generation of the new band formed at 590-650 nm (Fig. 4a) thus proving that the quenching is by the EnT mechanism.

The related discussion is also included in the revised manuscript on page 14.

Reviewer's Comment 10: A discussion why the mechanism is different from the mechanism of related reactions might be helpful (e.g. the work by Doyle, ref. 32).

Answer: The related discussion is given at the beginning of the revision. It clearly explains how our EnT mechanism is different from the Ni(III) mechanism proposed by Doyle.

The related discussion is also included in the revised manuscript on page 14.

Reviewer's Comment 11: Unfortunately, there are no page or line numbers but the section after Figure 3 contains several minor errors: "...at 500 nm expected ..." (should read "...expected at 500 nm..."), "...got completely..." (should read "...completely..."), "...energy gaps..." (should read "...a...energy gap..."), "...concluded..." (should read "...confirmed..."), "...have shown..." (should read "...showed...", twice), "...have the excited..." (should read "...have an excited...").

Answer: All the corrections suggested by the reviewer are made in the revised manuscript.

Reviewer's Comment 12: The KIE should also be determined from separate experiments but not only from competition experiments. As it stands, it remains unclear whether the C-H bond is cleaved in the turnover-limiting step or not (cf. ref. 48).

Answer: As suggested by the reviewer, KIE is also determined by using rate measurements (Fig. 4b). A KIE value of 1.2 (using rate measurement) and 1.7 (by competition experiments) reveal that the C-H bond breaking is not involved in the rate-determining step. The related discussion is also included in the revised manuscript on page 16.

We would like to thank all reviewers for their valuable comments which clearly improved the manuscript!

REVIEWER COMMENTS

Reviewer #3 (Remarks to the Author):

The authors have addressed in a convincing way the questions I have raised in my previous report. This work can now be accepted for publication in Nature Communications.

Reviewer #5 (Remarks to the Author):

In the manuscript at hand, Kancherla et al. report the alkylation of highly activated Csp³-H bonds with alkyl halides using dual nickel- and photocatalysis, along with extensive mechanistic investigations.

From a synthetic standpoint, the transformation clearly lacks any level of novelty required for a journal as Nat. Commun. Compared to the study by Paixao, König and co-workers (Adv. Synth. Catal. 2020), the disclosed synthetic method shows a largely identical substrate scope, while relying on significantly more expensive (Ir photocatalyst instead of an organic dye) or operationally demanding (Ni(cod)₂ instead of Ni(acac)₂) reagents and catalysts.

However, in my view, the major advance of this study is the highly detailed mechanistic analysis provided by the authors, combining spectroscopic and computational tools unravel the nature of the involved nickel intermediates and the role of the photocatalyst. The key finding is that all elementary steps of the nickel catalytic cycle (oxidative addition, Ni-halide bond homolysis, reductive elimination) proceed on the triplet hypersurface, whereas singlet intermediates represent off-cycle resting states: a finding that – if the hypothesis could be confirmed beyond doubt – would significantly advance the field of dual nickel-photoredox catalysis.

The authors provide extensive experimental evidence supporting their hypothesis, including

- the observation of a “triplet energy threshold” to obtain reactivity
- steady-state and time-resolved emission spectroscopy to prove quenching of the photocatalyst luminescence by an alkyl nickel(II) intermediate and to determine quenching kinetics

- transient absorption spectroscopy, showing the formation of a new, long-lived transient absorption feature, which is likely to be a triplet state

For these reasons, I would be supportive of publishing this manuscript in Nat. Commun., given that a number of issues and inconsistencies within the study could be resolved, and the focus of the manuscript is shifted towards the discussion and investigation of triplet nickel intermediates.

In one major argument, the authors claim that – in contrast to the work by Paixao, König and co-workers – the reaction does not proceed via a photoredox-mediated Ni(II)/Ni(III)/Ni(I) cycle, and use a range of oxidizing photocatalysts to validate their hypothesis. This experimental result, however, is not conclusive in my eyes, since reactivity will not only depend on the Ni(II)-Ni(III) oxidation, but also a following Ni(I)-Ni(0) reduction step. Paixao, König and co-workers use 4-CzIPN, which is often considered the “organic redox equivalent” to Ir(dF-CF₃-ppy)₂(dtbpy)⁺, with very similar redox potentials, but a significantly lower excited state energy. In order to underline their hypothesis, I strongly suggest the authors to perform the control experiment using 4-CzIPN as a photocatalyst.

Regarding the computational analysis, I was surprised to find that both Ni(0) and Ni(II) bipyridine complexes show a very low singlet-triplet gap of 4–5 kcal mol⁻¹ – which means that the triplet state for these intermediates is even thermally accessible. I think it would be very important to comment on this observation. Is this common for these (well-studied) types of nickel complexes? Is this an MLCT-type state that is so close in energy to the singlet ground state? In this case, the reactivity of the metal center could be described as a formal Ni(x+1) center. Moreover, the TDDFT computations seem puzzling to me. While the authors show that for intermediate DS, the triplet state DT is only 4.8 kcal mol⁻¹ above the singlet ground state, the lowest-lying triplet state seems to be much higher in the TDDFT computations. In my eyes, these aspects raise the question whether single-reference DFT is an appropriate computational methodology for this scenario, especially since there seem to be multiple electronic configurations with similar energies involved, and the potential energy surface seems to involve multiple state crossings.

Moreover, the main energy transfer to the Nickel(II) complexes should be elaborated in more detail: The authors claim that Dexter energy transfer populates the T₅ state of the nickel(II) intermediate, from which Ni–Br homolysis can occur. Do the authors hypothesize that energy transfer to T₁–T₄ is a competitive deactivation pathway, or are there other factors (electronic coupling, reorganization energies) favoring one of these processes over the other? And if the hypothesis of reactivity from the T₅ state is correct, how does this behavior relate to Kasha’s rule, stating that higher excited state lifetimes are usually so short that emission only occurs from the lowest state of a given multiplicity? Given that the authors propose a highly underrepresented mechanistic pattern, a discussion of these aspects would be beneficial to aid the adaption of the presented results.

Additional remarks:

- Page 3: „...cross-coupling reactions without using a photosensitizer (PS) is quite challenging since the triplet state of organometallic Ni(II) cannot be easily populated due to its low visible-light absorption cross-section“. This statement is an unnecessary and misleading generalization. Organometallic Ni(II) complexes do possess low-lying triplet MLCT states, which can be readily populated by visible light absorption – these complexes usually show bright colors. It could also be shown that these states can be used for catalysis applications (e.g. Scholes & Doyle, JACS 2018).
- The authors mention the possibility of a Förster-type energy transfer on multiple occasions. I am highly confused by this discussion: Organometallic iridium(III) complexes are very well known to exclusively react from their triplet excited state (and I am not aware of any TDAF-type behaviour of these compounds) – which fundamentally excludes FRET due to basic spin conservation rules. I suggest the authors to remove the discussion of FRET as a possible reaction pathway from the manuscript.
- Neither Scheme 1 nor the optimization table contain information about the emission maximum of the light source used. This should be included.
- What is the role of the ligand used by the authors (compared to the “standard” dtbbpy used in Nickel-photoredox catalysis)?
- The discussion of differing substrate reactivity (particularly for dioxane and toluene) is vague in my eyes and should be rephrased. Specific points to address: What is the effect of an oxygen atom in the beta position? Is this not mainly an inductive effect (i.e. making the C–H bond less hydridic and thereby less prone to abstraction by an electrophilic radical)? How does the stability of the benzylic radical relate to its diminished reactivity (especially given that the addition of a radical to Nickel is an extremely fast process, see fundamental works by Kochi).
- The authors compute the energy profile of deprotonation steps using inorganic bases in solution. To the best of my knowledge, computing the energies of such proton transfer steps (as well as the computation of accurate pKa values) represents a major source of error in computational chemistry, especially with ionic compounds (K₂CO₃, KBr) involved. How do the authors compute these values? I could not find any information in the Supplementary Information.
- The authors provide a rather speculative assessment of the transient absorption spectrum of the reduced Iridium(II) catalyst. If spectro-electrochemistry is available, this would be a good opportunity to prove this. Otherwise, I would encourage the authors to refer to literature where this has been performed (e.g. Glorius, Chem 2019).
- Did the authors determine the quantum yield of their transformation, or the dependence of the reaction rate on light intensity? Given that the proposed mechanism involves multiple energy transfer steps, this experiment could provide further insights into the mechanism.

Response to the reviewers

We would like to thank the reviewers for their detailed evaluation and the constructive comments! We revised the manuscript and supporting information as per the reviewer's and editor's suggestions. We have highlighted the changes with "yellow background" in the revised manuscript and the details of our response for the respective comments are given below.

Reviewer #3 (Remarks to the Author):

The authors have addressed in a convincing way the questions I have raised in my previous report. This work can now be accepted for publication in Nature Communications.

We thank the reviewer for accepting the manuscript for publication in Nature Communications.

Reviewer #5 (Remarks to the Author):

In the manuscript at hand, Kancherla et al. report the alkylation of highly activated Csp³-H bonds with alkyl halides using dual nickel- and photocatalysis, along with extensive mechanistic investigations.

From a synthetic standpoint, the transformation clearly lacks any level of novelty required for a journal as Nat. Commun. Compared to the study by Paixao, König and co-workers (Adv. Synth. Catal. 2020), the disclosed synthetic method shows a largely identical substrate scope, while relying on significantly more expensive (Ir photocatalyst instead of an organic dye) or operationally demanding (Ni(cod)₂ instead of Ni(acac)₂) reagents and catalysts.

However, in my view, the major advance of this study is the highly detailed mechanistic analysis provided by the authors, combining spectroscopic and computational tools unravel the nature of the involved nickel intermediates and the role of the photocatalyst. The key finding is that all elementary steps of the nickel catalytic cycle (oxidative addition, Ni-halide bond homolysis, reductive elimination) proceed on the triplet hypersurface, whereas singlet intermediates represent off-cycle resting states: a finding that – if the hypothesis could be confirmed beyond doubt – would significantly advance the field of dual nickel-photoredox catalysis.

The authors provide extensive experimental evidence supporting their hypothesis, including

- the observation of a “triplet energy threshold” to obtain reactivity.
- steady-state and time-resolved emission spectroscopy to prove quenching of the photocatalyst luminescence by an alkyl nickel(II) intermediate and to determine quenching kinetics.
- transient absorption spectroscopy, showing the formation of a new, long-lived transient absorption feature, which is likely to be a triplet state.

For these reasons, I would be supportive of publishing this manuscript in Nat. Commun., given that a number of issues and inconsistencies within the study could be resolved, and the focus of the manuscript is shifted towards the discussion and investigation of triplet nickel intermediates.

Reviewer's Comment 1: In one major argument, the authors claim that – in contrast to the work by Paixao, König and co-workers – the reaction does not proceed via a photoredox-mediated Ni(II)/Ni(III)/Ni(I) cycle, and use a range of oxidizing photocatalysts to validate their hypothesis. This experimental result, however, is not conclusive in my eyes, since reactivity will not only depend on the Ni(II)-Ni(III) oxidation, but also a following Ni(I)-Ni(0) reduction step. Paixao, König and co-workers use 4-CzIPN, which is often considered the “organic redox equivalent” to Ir(dF-CF₃-ppy)₂(dtbpy)⁺, with very similar redox potentials, but a significantly lower excited state energy. In order to underline their hypothesis, I strongly suggest the authors to perform the control experiment using 4-CzIPN as a photocatalyst.

Answer: Following the reviewer's suggestion, a reaction with 4-CzIPN as a photocatalyst was performed under the standard reaction conditions which gave the product in 30% yield. In the report by Paixao, König and co-workers, the EnT mechanism was ruled out as the reaction under UV irradiation in the absence of PC did not result in product formation.

The logic behind this reaction is that we should not expect product formation without using an external photocatalyst, since the absorptions (470 and 283 nm) of the Ni(II)-alkyl bromide intermediate **D^S** correspond to singlet a metal-to-ligand charge transfer (¹MLCT) and a ligand-centered π - π^* transition, respectively.

To realize the cross-coupling, one electron from an occupied molecular orbital should be promoted to the Ni–Br σ^* orbital, which triggers the Ni–Br bond breaking event. Analysis of the orbitals contributing to the dominant electronic transitions of **D^S** in the UV-Vis region revealed that all excitations correspond to transitions of electrons from the occupied Ni orbitals to the LUMO, LUMO+1, and LUMO+2, which represent π^* molecular orbitals located on the ligand (Supplementary Table 3, and Supplementary Fig. 16). These excitations are not effective in promoting the reactivity of the Ni(II) complex since the electron is not promoted to the Ni–Br σ^* orbital (LUMO+3). Analysis of the molecular orbitals of **D^S** indicates that LUMO+3, which essentially corresponds to the Ni–Br σ^* orbital (Supplementary Fig. 16), is involved in the **T⁵** excited state, ***D^{T5}**, with a singlet-triplet energy gap, $\Delta G_{T^5-S_0}$, of 56.2 kcal/mol. Therefore, to realize the product formation an external photocatalyst with energy greater than 56.2 kcal/mol is necessary (exergonic EnT).

4-CzIPN has potentials of $E_{1/2}(\text{PC}^*/\text{PC}^{\bullet-}) = +1.43\text{V}$; $E_{1/2}(\text{PC}/\text{PC}^{\bullet-}) = -1.24\text{V}$ vs SCE, and an excited-state triplet energy of $E_T = 58.3$ kcal/mol (*J. Am. Chem. Soc.* **2018**, *140*, 13719–13725). Interestingly, 4-CzIPN also have reasonably high triplet energy which can give the cross-coupled product.

Reviewer's Comment 2: Regarding the computational analysis, I was surprised to find that both Ni(0) and Ni(II) bipyridine complexes show a very low singlet-triplet gap of 4–5 kcal mol⁻¹ – which means that the triplet state for these intermediates is even thermally accessible. I think it would be very important to comment on this observation. Is this common for these (well-studied) types of nickel complexes? Is this an MLCT-type state that is so close in energy to the singlet ground state? In this case, the reactivity of the metal center could be described as a formal Ni(x+1) center. Moreover, the TDDFT computations seem puzzling to me. While the authors show that for intermediate DS, the triplet state DT is only 4.8 kcal mol⁻¹ above the singlet ground state, the lowest-lying triplet state seems to be much higher in the TDDFT computations. In my eyes, these aspects raise the question whether single-reference DFT is an appropriate computational methodology for this scenario, especially since there seem to be multiple electronic configurations with similar energies involved, and the potential energy surface seems to involve multiple state crossings.

Answer: We are grateful to the reviewer for constructive comments and suggestions. The triplet states of Ni(0) and Ni(II) reported in this study are ³d-d types, which is in agreement with the recent report (*J. Am. Chem. Soc.* **2020**, 142, 5800–5810). Although a very low singlet-triplet energy gap of 4-5 kcal/mol is found for both Ni(0) and Ni(II) bipyridine complexes (**A^S** and **D^S**), the ³d-d type of triplet is thermally inaccessible from the ground state. Photoexcitation of ground state singlet state leads to the ¹MLCT type excited state, which eventually gives the ³MLCT type triplet. The ³MLCT is a higher energy state, which eventually undergoes IC (internal conversion) to the most stable ³d-d state (*J. Am. Chem. Soc.* **2020**, 142, 5800–5810). However, in the current alkylation reaction the ³d-d state is unproductive.

In TDDFT calculations we desired to obtain electronic information of higher energy triplet states. The low lying triplets (T₁-T₄) in the TDDFT calculations are ³MLCT types (vertical excitation), whereas the **D^T** is ³d-d type (optimized geometry). Therefore the energy of **D^T** is not comparable with the TDDFT calculated triplet state.

The multireference calculations and other in-depth studies are underway in a separate project, where we will elaborately discuss the mechanism of energy transfer. In this current manuscript we mainly focus on the clear distinction between electron transfer (ET) and energy transfer (EnT) mechanism.

The related discussion is now given in the supplementary material (Supplementary Discussion XIII).

Reviewer's Comment 3: Moreover, the main energy transfer to the Nickel(II) complexes should be elaborated in more detail: The authors claim that Dexter energy transfer populates the T₅ state of the nickel(II) intermediate, from which Ni–Br homolysis can occur. Do the authors hypothesize that energy transfer to T₁–T₄ is a competitive deactivation pathway, or are there other factors (electronic coupling, reorganization energies) favoring one of these processes over the other? And if the hypothesis of reactivity from the T₅ state is correct, how does this behavior relate to Kasha's

rule, stating that higher excited state lifetimes are usually so short that emission only occurs from the lowest state of a given multiplicity? Given that the authors propose a highly underrepresented mechanistic pattern, a discussion of these aspects would be beneficial to aid the adaptation of the presented results.

Answer: The combined TDDFT calculations and NPA analysis suggest that low lying triple excited states T_1 – T_4 corresponds to ${}^3\text{MLCT}$. Therefore T_1 – T_4 states are not proficient for $\text{C}(sp^3)$ – H activation via Ni – Br homolysis. On the other hand T_5 state correspond to electron population at Ni – Br σ^* bond, which is mostly contributed by Ni – $d_{x^2-y^2}$ orbital. Therefore, the T_5 state is characterized as a ${}^3\text{d-d}$ type state which readily undergoes Ni – Br homolysis, and there will be a minor chance of IC (internal conversion) from T_5 to lower energy T_1 – T_4 states. Of course, the Dexter EnT transfer can occur from ${}^*\text{PS1}$ to any of the first 5 triplet excited states of D^{S} , T_n , $n=1$ – 5 , with only that leading to T_5 being effective in catalysis. However, we cannot consider that the population of T_1 – T_4 states are predicted as deactivation pathway (since the catalyst is not becoming inactive), they will convert to D^{S} via ISC (inter system crossing).

The related discussion is now given in the supplementary material (Supplementary Discussion XIII).

Additional remarks:

Reviewer's Comment 4: Page 3: „...cross-coupling reactions without using a photosensitizer (PS) is quite challenging since the triplet state of organometallic Ni(II) cannot be easily populated due to its low visible-light absorption cross-section”. This statement is an unnecessary and misleading generalization. Organometallic Ni(II) complexes do possess low-lying triplet MLCT states, which can be readily populated by visible light absorption – these complexes usually show bright colors. It could also be shown that these states can be used for catalysis applications (e.g. Scholes & Doyle, JACS 2018).

Answer: Following the reviewer's suggestion, the sentence is now rephrased as “Generally, in nickel catalysis [Ni(II)RX (R = aryl or alkyl; X = halogen)], achieving $\text{C}(sp^3)$ – $\text{C}(sp^2)$ and $\text{C}(sp^3)$ – $\text{C}(sp^3)$ cross-coupling reactions without using a photosensitizer (PS) is rather challenging since the Ni(II) – Br σ^* orbital cannot be easily populated by the direct irradiation of nickel complexes.”

Reviewer's Comment 5: The authors mention the possibility of a Förster-type energy transfer on multiple occasions. I am highly confused by this discussion: Organometallic iridium(III) complexes are very well known to exclusively react from their triplet excited state (and I am not aware of any TDAF-type behaviour of these compounds) – which fundamentally excludes FRET due to basic spin conservation rules. I suggest the authors to remove the discussion of FRET as a possible reaction pathway from the manuscript.

Answer: Following the reviewer's suggestion, we now limited the discussion of FRET as a possible reaction pathway in the revised manuscript.

Reviewer's Comment 6: Neither Scheme 1 nor the optimization table contain information about the emission maximum of the light source used. This should be included.

Answer: We used Kessil H150, 34W blue LEDs with emission maximum at 425 nm for performing the reactions. The emission maximum of the light source is now given in Table 1 and in the supporting material.

Reviewer's Comment 7: What is the role of the ligand used by the authors (compared to the “standard” dtbbpy used in Nickel-photoredox catalysis)?

Answer: Although dtbbpy is used as a standard ligand for most of the photoredox Ni-dual catalysis, the use of dOMe-bpy ligand is not unusual. We did not perform an extensive catalyst-ligand screening. According to the optimization data, the use of dOMe-bpy ligand gave a better yield compared to dtbbpy ligand. Interestingly, the MacMillan group also used dOMe-bpy as a ligand for C(sp³)-C(sp³) cross-coupling of alkyl bromides with carboxylic acids, α-C-H of amines, ethers and sulphides [*Nature* **536**, 322-325 (2016); *Nature* **547**, 79-83 (2017)].

Reviewer's Comment 8: The discussion of differing substrate reactivity (particularly for dioxane and toluene) is vague in my eyes and should be rephrased.

Specific points to address: What is the effect of an oxygen atom in the beta position? Is this not mainly an inductive effect (i.e. making the C-H bond less hydridic and thereby less prone to abstraction by an electrophilic radical)? How does the stability of the benzylic radical relate to its diminished reactivity (especially given that the addition of a radical to Nickel is an extremely fast process, see fundamental works by Kochi).

Answer: Following the reviewer's suggestion, the discussion regarding the substrate reactivity is rephrased.

“In the case of 1,4-dioxane, the diminished reactivity might be mainly due to inductive effect making the C-H bond less hydridic and there by less prone to abstraction by the electrophilic radical.^{38,39} Additionally, as non-ethereal substrate toluene was to be found reactive and gave the cross-coupled product **28**. In this case an outer-sphere C(sp³)-H activation may be operative which is giving a stable benzylic radical. Once the benzylic radical is formed it can either attack the alkyl-Ni(I) intermediate (E^D, ΔG = -15.8 kcal/mol) or dimerizes (ΔG = -24.4 kcal/mol). Since the dimerization of the benzylic radical is more favored over the addition to alkyl-Ni(I) intermediate (E^D), the reaction with toluene resulted in a lower yield compared to THF. We also tried to use THF in equivalent amounts, however, homocoupling of alkyl halide was obtained as a major product. Therefore, we used THF as solvent in order to suppress the homocoupling and to promote the cross-coupling reaction.”

Reviewer's Comment 9: The authors compute the energy profile of deprotonation steps using inorganic bases in solution. To the best of my knowledge, computing the energies of such proton transfer steps (as well as the computation of accurate pKa values) represents a major source of error in computational chemistry, especially with ionic compounds (K₂CO₃, KBr) involved. How do the authors compute these values? I could not find any information in the Supplementary Information.

Answer: We completely agree with the reviewer that the deprotonation step is the source of error in computational chemistry. However, in this mechanism (Fig. 2a) the C(sp³)-H activation occurs via hydrogen atom transfer (not proton transfer) by an active bromine atom to form HBr. This is not a deprotonation step, and hence we have not taken care about the computational error.

Reviewer's Comment 10: The authors provide a rather speculative assessment of the transient absorption spectrum of the reduced Iridium(II) catalyst. If spectro-electrochemistry is available, this would be a good opportunity to prove this. Otherwise, I would encourage the authors to refer to literature where this has been performed (e.g. Glorius, Chem 2019).

Answer: We thank the reviewer for this important comment. We have referred the work by Glorius and coworkers (Chem 5, 2183-2194 (2019)), who showed the transient absorption features of reduced **PS1** (i.e. PS1^{•-}). The following discussion is now included in the revised manuscript by referring to their paper.

“In case for the reaction to proceed via a Ni(III) intermediate by adding the quencher (**D^S**) to ^{3*}**PS1**, an electron from Ni(II) enters into the empty t_{2g} orbital of ^{3*}**PS1** which results in the formation of reduced **PS1** (i.e. PS1^{•-}) with new transient spectral features and maxima at 400, 443, 499, and 530 nm.⁵² Therefore, with the increase of the quencher (**D^S**) concentration, the ESA peaks (470 and 850 nm) corresponding to ^{3*}**PS1** should decay, and new TA signals corresponding to **PS1^{•-}** and Ni(III) should appear. Overall, the quenching rate for these TA signals should be different for the ET pathway.⁵³ However, Fig. 3d-g, and Supplementary Fig. 8 clearly show that ESA peaks at 470 and 850 nm have disappeared with the concentration increase of **D^S** and new transient peaks (ESA and GSB) appeared that are different from **PS1^{•-}**. Moreover, the quenching rate (450-480, and 775-880 nm) and the formation rate (590-650 nm) of the transient signals are identical (Fig. 4a) which clearly shows that the ET mechanism is not operative and an alternative EnT pathway is taking place. In the EnT pathway, spectroscopically, mutual exchange of electrons between ^{3*}**PS1** and **D^S** should result in the decay of ESA transient signals (470 and 850 nm, corresponding to reduced bipyridine ligand of ^{3*}**PS1**), since it transfers that electron to the T⁵ excited state of Ni(II). Simultaneously an electron from the ground state **D^S** will be promoted to empty t_{2g} orbital of ^{3*}**PS1** giving rise to ground-state **PS1** and excited ***D^T**. Therefore the overall transient spectral changes involve the decay of ESA (470 and 850 nm), and the appearance of new transient signals corresponding to the excited ***D^T**. In addition, the excited ***D^T** should have a long lifetime as the mutual exchange of electrons results in the formation of a spin-flipped state. As expected, the ns-TA of ***D^T** displayed a long-lived excited state with a weighted average lifetime of $\tau = 671$ ns

(Supplementary Fig. 11) and we assign this newly formed long-lived excited state of nickel to be a spin-flipped triplet state. Besides $\text{PSI}^{\bullet-}$ has a weighted average life time of $\tau = 86.71 \mu\text{s}$ which is much higher than the life time of the new transient species ($\tau = 671 \text{ ns}$) observed in our system, which further confirms that $\text{PSI}^{\bullet-}$ is not forming in our system.⁵² Also, the TA bands at 450-480, and 775-880 nm were found to decay at the same rate with the generation of the new band formed at 590-650 nm (Fig. 4a) thus supporting that the quenching is occurring by the EnT mechanism.”

Reviewer’s Comment 11: Did the authors determine the quantum yield of their transformation, or the dependence of the reaction rate on light intensity? Given that the proposed mechanism involves multiple energy transfer steps, this experiment could provide further insights into the mechanism.

Answer: Following the reviewer's suggestion, dependence of the reaction rate on light intensity is calculated. We have conducted the reactions under full and half intensity of the light source at different time intervals and the corresponding yields are measured by GCMS using dodecane as internal standard. The values suggest that more than one photon is involved in the reaction mechanism.

Time (h)	At full intensity, product/standard ratio	At half intensity, product/standard ratio
1	0	0
2	0.0271	0.01
4	0.0913	0.029
6	0.187	0.057
8	0.279	0.095

We would like to thank all reviewers specifically Reviewer-5 for the valuable comments which clearly improved the manuscript!

REVIEWERS' COMMENTS

Reviewer #5 (Remarks to the Author):

The authors have made substantial efforts to revise the manuscript and addressed all of my concerns and suggestions in a thoughtful and detailed manner. Therefore, I am happy to recommend this work for publication in Nature Communications. I want to congratulate the authors on this detailed and thorough mechanistic analysis. I feel that this – together with the recent studies by Doyle et al. – could take the understanding of dual nickel-photocatalytic processes to a new level.

To very minor remarks:

- I appreciate the detailed discussion on the nature of electronic states, especially the d–d nature of intermediates AT, DT FT (in agreement with the previous work by Doyle and co-workers), in contrast to the MLCT states obtained upon direct excitation. I feel that highlighting these aspects within the main text of the manuscript could contribute to a better understanding of the overall reactivity.

- The authors clearly explained their reasoning regarding the omission of the inorganic base within their computation (although this probably neglects a significant thermodynamic driving force for this elementary step). Yet, the potential energy curve shows a step that seems to involve the conversion of K_2CO_3 and HBr to $KHCO_3$ and KBr. I would recommend the authors to remove this from Fig. 2 (and the corresponding figures in the Supplementary Information) to avoid confusion.

Response to the reviewers

We would like to thank the reviewer for the detailed evaluation and the constructive comments! We revised the manuscript and supporting information as per the reviewer's and editor's suggestions. We have highlighted the changes with "yellow background" in the revised manuscript and the details of our response for the respective comments are given below.

Reviewer #5 (Remarks to the Author):

The authors have made substantial efforts to revise the manuscript and addressed all of my concerns and suggestions in a thoughtful and detailed manner. Therefore, I am happy to recommend this work for publication in Nature Communications. I want to congratulate the authors on this detailed and thorough mechanistic analysis. I feel that this – together with the recent studies by Doyle et al. – could take the understanding of dual nickel-photocatalytic processes to a new level.

We thank the reviewer for accepting the manuscript for publication in Nature Communications.

Reviewer's Comment 1: I appreciate the detailed discussion on the nature of electronic states, especially the d–d nature of intermediates AT, DT FT (in agreement with the previous work by Doyle and co-workers), in contrast to the MLCT states obtained upon direct excitation. I feel that highlighting these aspects within the main text of the manuscript could contribute to a better understanding of the overall reactivity.

Answer: Following the reviewer's suggestion, the discussion on the nature of electronic states is now highlighted in the revised manuscript. The following discussion highlighting these aspects can be found in the revised manuscript.

“To investigate the above possible pathway and to obtain electronic information of higher energy triplet states, we performed TD-DFT calculations (Supplementary Methods 14). Analysis of the orbitals contributing to the dominant electronic transitions of \mathbf{D}^S in the UV-Vis region revealed that all the excitations correspond to transitions of electrons from occupied Ni orbitals to the LUMO, LUMO+1, and LUMO+2, which represent π^* molecular orbitals located on the dOMe-bpy ligand (Supplementary Table 5, and Supplementary Fig. 16). These excitations are not effective in promoting the reactivity of the Ni(II) complex towards bromine radical formation since the electron is not promoted to the Ni–Br σ^* orbital (LUMO+3), as this would be a Laporte-forbidden d–d* transition. Confirming this finding experimentally, (dtbbpy)Ni(II)-alkyl bromide prepared in situ displayed broad absorption features with λ_{\max} values of 470 and 283 nm due to metal-to-ligand charge transfer ($^1\text{MLCT}$) and a ligand-centered $\pi \rightarrow \pi^*$ transition, respectively (Supplementary Fig. 1). An ISC (inter-system crossing) of direct photoexcited state $^1\text{MLCT}$ leads to $^3\text{MLCT}$ type triplet, which eventually undergoes IC (internal conversion) to the most stable $^3\text{d-d}$ state \mathbf{D}^T which is in agreement to the recent report by Doyle and coworkers. However, in the current C(sp^3)–H alkylation reaction the $^3\text{d-d}$ state \mathbf{D}^T is unproductive, since it is not the source of bromide radical. Irradiating the in situ-generated Ni(II)-alkyl bromide complex with visible and

ultraviolet light in the absence of **PS1** either resulted in no cross-coupled product (with visible light) or trace product (with UV light, 300 nm), supporting the above conclusions.”

Reviewer’s Comment 2: The authors clearly explained their reasoning regarding the omission of the inorganic base within their computation (although this probably neglects a significant thermodynamic driving force for this elementary step). Yet, the potential energy curve shows a step that seems to involve the conversion of K_2CO_3 and HBr to $KHCO_3$ and KBr. I would recommend the authors to remove this from Fig. 2 (and the corresponding figures in the Supplementary Information) to avoid confusion.

Answer: Thank you for the suggestion. In our mechanism (Fig. 4a) the $C(sp^3)$ -H activation occurs via hydrogen atom transfer by an active bromine atom to form HBr which then reacts to form KBr. The energy difference of $\Delta G = -32.1$ kcal/mol from $[E-F]^{\ddagger}$ to F^{\ddagger} also includes the conversion of K_2CO_3 to $KHCO_3$ and KBr. Therefore, to avoid the confusion that could arise between HAT and a proton transfer process, we included the sentence “The HAT step leads to the formation of HBr which is readily transformed into KBr and $KHCO_3$ in presence of K_2CO_3 ” in the Fig. 4 legend.

We would like to thank all reviewers for the valuable comments which clearly improved the manuscript!